# Dual-Force: Enhanced Offline Diversity Maximization under Imitation Constraints

## Abstract

While many algorithms for diversity maximization under imitation constraints are online in nature, many applications require offline algorithms without environment interactions. Tackling this problem in the offline setting, however, presents significant challenges that require non-trivial, multi-stage optimization processes with non-stationary rewards. In this work, we present a novel offline algorithm that enhances diversity using an objective based on Van der Waals (VdW) force and successor features, and eliminates the need to learn a previously used skill discriminator. Moreover, by conditioning the value function and policy on a pre-trained Functional Reward Encoding (FRE), our method allows for better handling of non-stationary rewards and provides zero-shot recall of all skills encountered during training, significantly expanding the set of skills learned in prior work. Consequently, our algorithm benefits from receiving a consistently strong diversity signal (VdW), and enjoys more stable and efficient training. We demonstrate the effectiveness of our method in generating diverse skills for two robotic tasks in simulation: locomotion of a quadruped and local navigation with obstacle traversal.[1]

## 1 Introduction

Leveraging demonstration data has established itself as one of the main directions for large-scale learning systems. This is due to the abundance and ubiquity of demonstration data from various sources, such as videos, robots, and more. There are several arguments as to why we should not stop at naive learning from demonstrations. First, they are often not ego-centric and come externally to the agent, meaning that the state space of the demonstration needs to be matched to the agent. Second, the agent may not be able to fully replicate the demonstration due to limited capabilities (Li et al., 2023). This suggests that an agent must necessarily adapt the demonstration to its capabilities, which is achieved by extracting diverse behaviors that are close to the demonstration (Vlastelica et al., 2024). Moreover, another important aspect is robustness to distribution shifts. Tasks may be solved in various ways, some are more robust than others. Extracting diverse policies enables us to choose the more robust alternatives (Vlastelica et al., 2024). Alternatively, if we can quantify the risk of acting with a particular policy, we can encourage risk-averse behavior, which is an orthogonal approach (Vlastelica et al., 2022).

Previous work on maximizing diversity under various constraints has been formalized in the *Constrained Markov Decision Process* formulation (Zahavy et al., 2023; Vlastelica et al., 2024; Cheng et al., 2024). Solving the underlying constraint optimization problem involves a Lagrangian relaxation (a two-phase alternating scheme) in which the constraints are lifted to the (reward) objective and scaled by Lagrangian multipliers that adaptively reduce constraint violations (Zahavy et al., 2023; Cheng et al., 2024). In contrast to the online setting (Zahavy et al., 2023; Cheng et al., 2024), we focus here on the offline setting without environment interaction and relax constraints that enforce near-optimal returns by considering imitation constraints, as in Vlastelica et al. (2024). Our approach to offline learning from demonstrations crucially relies on the Fenchel duality theory adapted to the Reinforcement Learning (RL) setting in the DIstribution Correction Estimation (DICE) framework (Nachum and Dai, 2020; Kim et al., 2022; Ma et al., 2022a;b). Prior work by Vlastelica

---

[1]Project website with videos: https://tinyurl.com/dual-force

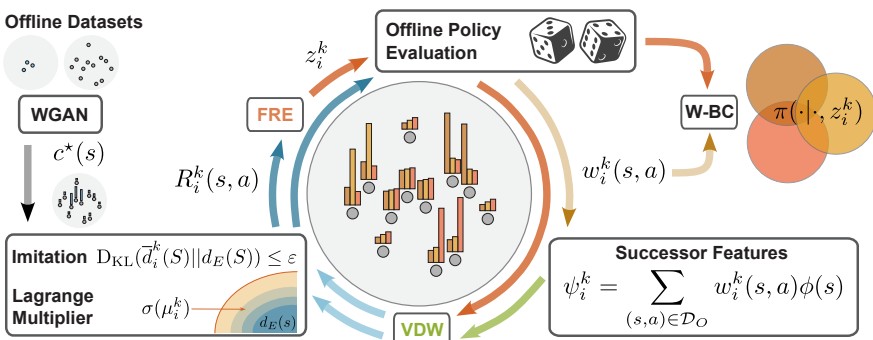

Figure 1: Illustration of Dual-Force. The pseudocode is presented in Algorithm 1.

et al. (2024) considers diversity objective based on a variational lower bound on mutual information between states and skills, which inevitably leads to learning a skill-discriminator. In addition to introducing another training phase into the alternating scheme, this design choice faces several practical challenges: i) a single-step policy and skill discriminator update in the offline setting does not provide as accurate a policy estimate as sampling a Monte Carlo trajectory in the online setting (Eysenbach et al., 2019); ii) this inaccuracy combined with the non-stationary reward (Lagrange multipliers and skill-discriminator) results in a skill-discriminator that fails to accurately discriminate skills; and iii) while this can be alleviated by introducing an additional information gain term (Strouse et al., 2022) into the objective, its effect can quickly vanish in the offline setting. An important open problem posed by (Vlastelica et al., 2024) is overcoming the difficulty of training the skill-discriminator, which is a significant challenge in practice. Furthermore, their algorithm (DOI) violates the stationary reward assumption in the DICE framework, making the training phase of the value function potentially unstable. In addition, prior work by (Zahavy et al., 2023) and (Vlastelica et al., 2024) is limited by the requirement to predefine the number of skills, resulting in learning a policy that captures only the most recent skills while forgetting previously learned ones, and has a runtime complexity that is linear in the number of skills.

In this work, we present Dual-Force, a novel offline algorithm that addresses the previous challenges. The crux of our approach is to give an off-policy evaluation procedure for a physically inspired diversity objective. In particular, we achieve enhanced diversity using the Van der Waals (VdW) force (Zahavy et al., 2023) which allows us to eliminate the need to learn the skill discriminator in (Vlastelica et al., 2024) and provides us with a strong diversity signal in the offline setting and even throughout the whole execution. Our key technical insight is that all relevant quantities needed to compute the VdW force, including successor features (Dayan, 1993; Barreto et al., 2017) and dual-conjugate variables (see Sec. 3), can be estimated off-policy using an importance sampling technique from the DICE framework. Furthermore, we enable value function (and policy) training in the non-stationary setting by conditioning it on a pre-trained Functional Reward Encoding (FRE) (Frans et al., 2024) that maps each skill reward to a compact latent representation, in each phase. In addition, for each non-stationary reward encountered during training, we can recall the corresponding skill by its associated latent FRE embedding. At a minimal cost of storing these latent FRE embeddings, our algorithm significantly expands the set of diverse skills learned (e.g., prior work learns only five skills), as their number scales proportionally with the number of phases performed. We demonstrate the effectiveness of our method on two offline datasets collected from a 12-DoF quadruped robot Solo12 (Vlastelica et al., 2024). Specifically, we show that Dual-Force can efficiently and robustly recover diverse behaviors in an offline dataset, all of which imitate a target expert state occupancy[2].

## 2 Preliminaries

We utilize the framework of Markov decision processes (MDPs) (Puterman, 2014), where an MDP is defined by the tuple $(\mathcal{S}, \mathcal{A}, \mathcal{R}, \mathcal{P}, \rho_0, \gamma)$ denoting the state space, action space, reward mapping $\mathcal{R} : \mathcal{S} \times \mathcal{A} \mapsto \mathbb{R}$, stochastic

---

[2]Similarly to previous work (Ma et al., 2022b), our results are generalizable to the setting of state-occupancy imitation constraints specified by an arbitrary $f$-divergence measure.

transition kernel $\mathcal{P}(s'|s,a)$, initial state distribution $\rho_0(s)$ and discount factor $\gamma$. A policy $\pi : \mathcal{S} \mapsto \Delta(\mathcal{A})$ defines a probability distribution over the action space $\mathcal{A}$ conditioned on a state, where $\Delta(\cdot)$ stands for the probability simplex.

Given a policy $\pi$, the corresponding state-action occupancy measure is defined by

$$d_\pi(s,a) := (1-\gamma) \sum_{t=0}^{\infty} \gamma^t \Pr[s_t = s, a_t = a \mid s_0 \sim \rho_0, a_t \sim \pi(\cdot|s_t), s_{t+1} \sim \mathcal{P}(\cdot|s_t, a_t)]$$

and its associated state occupancy $d_\pi(s)$ is given by marginalizing over the action space $\sum_{a \in \mathcal{A}} d_\pi(s,a)$. The RL objective can be rewritten as maximizing a function of the occupancy measure $\max_{d_\pi \in \mathcal{K}} \langle d_\pi, r \rangle$, where $\langle d_\pi, r \rangle = \sum_{s,a} d_\pi(s,a) r(s,a)$ denotes the inner product and $\mathcal{K}$ is the set of admissible distributions (Zahavy et al., 2021). We will consider a diversity objective with input $n$ state-action occupancies $(d_1, \ldots, d_n)$, where $d_i$ is induced by a policy $\pi_i$.

We consider an offline setting with access to the following datasets: i) $\mathcal{D}_E$ sampled from an expert state occupancy $d_E(S)$; and ii) $\mathcal{D}_O$ sampled from a state-action occupancy $d_O(S, A)$ generated by a mixture of behaviors.

## 2.1 Constrained Markov Decision Process (CMDP)

Zahavy et al. (2023) studied a CMDP formulation that seeks to compute a set of policies $\Pi^n = \{\pi_i\}_{i=1}^n$ that satisfy

$$\max_{\Pi^n} \text{ Diversity}(\Pi^n) \text{ subject to } \langle d_\pi, r_e \rangle \geq \alpha v_e^*, \quad \forall \pi \in \Pi^n, \tag{1}$$

where $r_e$ is an extrinsic reward and $v_e^*$ an optimal extrinsic value. Intuitively, eq. (1) computes a set of diverse policies while maintaining a certain level of extrinsic optimality specified by a parameter $\alpha \in (0, 1]$. They designed a heuristic for optimizing convex diversity objectives by solving a sequence of standard RL problems, each with an intrinsic reward equal to the gradient of the objective evaluated at the previous step (say $k^{\text{th}}$) time-averaged state-action occupancies $\{\overline{d}_1^k, \ldots, \overline{d}_n^k\}$, namely

$$r_i^{k+1} = \nabla_{d_i} \text{Diversity}(\overline{d}_1^k, \ldots, \overline{d}_n^k), \quad \forall i \in \{1, \ldots, n\}. \tag{2}$$

The Lagrange relaxation of the CMDP in eq. (1) becomes an RL problem with a reward function that depends on Lagrange multiplier $\lambda \geq 0$ that balances the extrinsic and intrinsic reward (Borkar, 2005)

$$r^{k+1} = r_e + \lambda_i r_i^{k+1}, \quad \forall z, \tag{3}$$

where the Lagrange multipliers are minimizing the following loss

$$\mathcal{L}_\lambda = \sum_{i=1}^n \lambda_i(\langle d_i, r_e \rangle - \alpha v_e^*). \tag{4}$$

Intuitively, a Lagrange multiplier increases when the associated constraint is violated and decreases otherwise. The practical implementation considers an extrinsic and intrinsic advantage coupled with bounded Lagrange multipliers (Stooke et al., 2020; Cheng et al., 2024), i.e., applying Sigmoid activation $\sigma(\mu_i)$ to an unbounded variable $\mu_i \in \mathbb{R}$, for all $i \in \{1, \ldots, n\}$.

## 2.2 Functional Reward Encoding (FRE)

Recently, Frans et al. (2024) proposed an information bottleneck method (Tishby et al., 2000; Alemi et al., 2017) for encoding state-reward samples into a latent representation using a transformer-based variational auto-encoder. Specifically, for a reward function $r$ sampled from a prior probability distribution, the latent representation $z_r$ encoded from an arbitrary subset of state-reward samples $L_r^{\text{enc}} = \{(s, r(s)) : s \in D_{\text{enc}}\}$ should be as compressive as possible, while being maximally predictive for decoding the rewards of other arbitrary subset of state-reward samples $L_r^{\text{dec}}$. The empirical evaluation demonstrated, in standard D4RL offline environments (Fu et al., 2020), that optimizing a latent-conditioned policy $\pi(\cdot|\cdot, z)$ with fixed Functional Reward Encoding (FRE) pretrained on diverse class of reward functions (random linear, MLP, and goal-reaching), enables zero-shot solving of downstream tasks. Namely, given arbitrary state-reward samples $L_r^{\text{enc}}$, the latent-conditioned policy $\pi(\cdot|\cdot, z_r)$ with latent representation $z_r = \text{FRE}(L_r^{enc})$ optimizes the reward $r$.

## 3 Problem Formulation

We aim to solve the following optimization problem,

$$\max_{d_1,\ldots,d_n} \quad \text{Diversity}(d_1,\ldots,d_n) \tag{5}$$

$$\text{subject to} \quad \text{D}_{\text{KL}}\left(d_i(S)||d_E(S)\right) \leq \varepsilon \quad \forall i \in \{1,\ldots,n\}, \tag{6}$$

where $d_E(S)$ is a state-only expert occupancy. This puts us in a similar setting as Vlastelica et al. (2024), with two key differences: (i) we shall introduce a more stable diversity objective; and (ii) we shall relax the state occupancy constraints while preserving their state-only occupancy nature, allowing for more freedom in diversity maximization.

### 3.1 Diversity Measures

Vlastelica et al. (2024) used a variational lower bound on a mutual information $\mathcal{I}(S; Z)$ between states and latent skills, resulting in the following diversity objective

$$\mathcal{I}(S; Z) \geq \mathbb{E}_{p(z), d_z(s)}\left[\log q(z|s)\right] + \mathcal{H}\left(p(z)\right) = \sum_{z \in Z} \mathbb{E}_{d_z(s)}\left[\frac{\log\left(|Z|q(z|s)\right)}{|Z|}\right], \tag{7}$$

where $p(z)$ is a categorical distribution over a discrete set $Z$ of $|Z|$ many distinct indicator vectors in $\mathbb{R}^{|Z|}$ and $d_z(s) := d_{\pi_z}(s)$ is a state occupancy induced by a skill-conditioned policy $\pi_z$. While they showed that this objective can be estimated off-policy using a DICE importance sampling approach, this comes at the cost of learning a skill-discriminator and makes the training unstable.

Zahavy et al. (2023) modeled a diversity objective, based on a distance measure in (Abbeel and Ng, 2004), as a maximization of a minimum squared $\ell_2$ distance between successor features of different skills, namely

$$\max_{d_1,\ldots,d_n} \quad \frac{1}{n}\sum_{i=1}^{n} \min_{j \neq i} \left\|\psi^i - \psi^j\right\|_2^2. \tag{8}$$

More specifically, given a feature mapping $\phi : \mathcal{S} \to \mathbb{R}^n$, the successor features (Dayan, 1993; Barreto et al., 2017) are defined by $\psi_i = \mathbb{E}_{d_i(s)}[\phi(s)]$. An important property of this convex objective is that its gradient is given in closed form, derived for completeness in Thm. C.1, eliminating the need to learn a skill discriminator.

Furthermore, Zahavy et al. (2023) introduced a physically inspired objective based on Van der Waals (VdW) force, and considered the following optimization objective

$$\max_{d_1,\ldots,d_n} \quad 0.5\sum_{i=1}^{n} \ell_i^2 - 0.2(\ell_i^5/\ell_0^3), \tag{9}$$

where $\ell_i := \left\|\psi_i - \psi_{j_i^\star}\right\|_2$ and $j_i^\star := \arg\min_{j \neq i} \left\|\psi_i - \psi_j\right\|_2$. In this work, we use this formulation as it allows the level of diversity to be controlled by a parameter $\ell_0$. When the successor features are in close proximity $\ell_i < \ell_0$, the repulsive force dominates, whereas when $\ell_i > \ell_0$ the attractive force prevails. In the setting when $\ell_0 \to \infty$, the formulation is eq. (8) is recovered.

## 4 Method

### 4.1 Van der Waals Force

Our key technical insight is that in the context of KL-divergence imitation constraints, see Sec. 3, all relevant quantities in the approach of (Zahavy et al., 2023), including dual-conjugate variables and successor features, can be estimated off-policy using a DICE importance sampling procedure.

From now on, in Problem 5, we set the diversity objective with the VdW force in eq. (9). Our first observation, formalized in Thm. A.2, is that the imitation constraints in Sec. 3 can be relaxed to

$$-\mathbb{E}_{d_i(s)}\left[\log\frac{d_E(s)}{d_O(s)}\right] + \mathrm{D_{KL}}\left(d_i(S,A)||d_O(S,A)\right) \leq \varepsilon, \qquad \forall i \in \{1,\ldots,n\}. \tag{10}$$

In this way, we use a tighter relaxation of the imitation constraints that preserves the state-occupancy nature and still admits efficient computation, instead of enforcing the more restrictive state-action occupancy constraints with respect to a fixed SMODICE-expert (Vlastelica et al., 2024).

Using similar arguments as in Zahavy et al. (2023), we arrive at an iterative procedure which in iteration $k+1$ considers the following Lagrange relaxation of Problem 5 for the $i^{\mathrm{th}}$ state-action distribution $d_i$:

$$\min_{\lambda_i \geq 0} \max_{d_i} \mathbb{E}_{d_i(s,a)}\left[\beta_i^k(s,a)\right] + \lambda_i\left[\mathbb{E}_{d_i(s,a)}\left[\log\frac{d_E(s)}{d_O(s)}\right] - \mathrm{D_{KL}}\left(d_i(S,A)||d_O(S,A)\right)\right], \tag{11}$$

where $\lambda_i$ is a Lagrange multiplier and $\beta_i^k = \nabla_{d_i}\mathrm{Diversity}(\overline{d}_1^k,\ldots,\overline{d}_n^k)$ is a dual-conjugate variable, which in our setting with diversity objective set to the VdW force in eq. (9), reduces to

$$\beta_i^k(s,a) := (1 - (\ell_i^k/\ell_0)^3)\langle\phi(s), \psi_i^k - \psi_{j^\star}^k\rangle,$$

where $\psi_i^k := \mathbb{E}_{\overline{d}_i^k(s)}[\phi(s)]$, $\ell_i^k := \|\psi_i^k - \psi_{j^\star}^k\|_2$ and $j^\star := \arg\min_{j\neq i}\left\|\psi_i^k - \psi_j^k\right\|_2$ are defined with respect to a time-averaged state-action distribution $\overline{d}_i^k = \frac{1}{t}\sum_{t=1}^k d_i^t$.

Next, we apply Fenchel duality to solve offline the inner maximization problem in eq. (11). Due to practical considerations, we use bounded Lagrange multipliers $\sigma(\mu_i)$ and Polyak update scheme to maintain the time-averaged state-action distributions $\{\overline{d}_1^k,\ldots,\overline{d}_n^k\}$. In particular, after fixing the bounded Lagrange multipliers, we obtain a standard RL problem regularized with a KL-divergence term

$$\max_{d_i} \mathbb{E}_{d_i(s,a)}\left[R_i^\mu(s,a)\right] - \mathrm{D_{KL}}\left(d_i(S,A)||d_O(S,A)\right), \tag{12}$$

where the non-stationary reward is given by

$$R_i^\mu(s,a) := \underbrace{(1 - \sigma(\mu_i))}_{\text{Constraint Satisfaction}}\underbrace{\beta_i^k(s,a)}_{\text{VdW-Diversity}} + \underbrace{\sigma(\mu_i)}_{\text{Constraint Violation}}\underbrace{\log\frac{c^\star(s)}{1 - c^\star(s)}}_{\text{Expert-Imitation}}. \tag{13}$$

Here, $c^\star(s)$ denotes a pretrained state-discriminator (Kim et al., 2022; Ma et al., 2022a) which distinguishes between the states in an expert dataset $\mathcal{D}_E \sim d_E(S)$ from the states in an offline dataset $\mathcal{D}_O \sim d_O(S,A)$, and satisfies $c^\star(s) = d_E(s)/(d_E(s) + d_O(s))$.

### 4.2 Offline Estimators using Fenchel Duality

The DICE framework (Nachum and Dai, 2020; Kim et al., 2022; Ma et al., 2022a;b) solves offline the KL-regularized RL problem in eq. (12) by considering its dual formulation, which reads

$$V_i^\star = \arg\min_{V(s)}(1 - \gamma)\mathbb{E}_{s\sim\rho_0}\left[V(s)\right] + \log\mathbb{E}_{d_O(s,a)}\exp\left\{R_i^\mu(s,a) + \gamma\mathcal{T}V(s,a) - V(s)\right\}, \tag{14}$$

where we denote by $\mathcal{T}V(s,a) := \mathbb{E}_{\mathcal{P}(s'|s,a)}V(s')$. The temporal difference (TD) error is given by

$$\delta_i(s,a) = R_i^\mu(s,a) + \gamma\mathcal{T}V_i^\star(s,a) - V_i^\star(s).$$

Then, the primal solution of Problem 11 is given by

$$\eta_i(s,a) := \frac{d_i^\star(s,a)}{d_O(s,a)} = \mathrm{softmax}_{d_O(s,a)}\left(\delta_i(s,a)\right) = \frac{\exp\{\delta_i(s,a)\}}{\mathbb{E}_{d_O(s',a')}\exp\{\delta_i(s',a')\}}. \tag{15}$$

Based on the importance ratios $\eta_i$ we can compute offline all necessary estimators. In particular, for any function $f$, we can estimate offline the following expectation:

$$\mathbb{E}_{d_i(s,a)}[f(s,a)] = \mathbb{E}_{d_O(s,a)}[\eta_i(s,a)f(s,a)]. \tag{16}$$

Using eq. (16) we can train offline an optimal policy by maximizing the following weighted behavior cloning objective $\mathbb{E}_{d_O(s,a)}[\eta_i(s,a)\log \pi_i(a|s)]$. Similarly, we can estimate offline the successor features $\psi_i = \mathbb{E}_{d_O(s,a)}[\eta_i(s,a)\phi(s,a)]$ and also maintain the associated averaged over time successor representations $\psi_i^k$. This gives us the tool to estimate offline the VdW-Diversity term in eq. (13).

Next, we dynamically adjust the bounded Lagrange multipliers $\sigma(\mu_i)$ based on an offline estimation of the corresponding constraint violation. In Theorem A.3, we show that the LHS of eq. (10) admits an estimator

$$\mathbb{E}_{d_O(s,a)}\left[\eta_i(s,a)\left(\log \eta_i(s,a) - \log \frac{c^\star(s)}{1-c^\star(s)}\right)\right]. \tag{17}$$

In practice, however, we only have access to finitely many samples of the state occupancy $d_O(s,a)$. Thus, in Theorem B.2, we derive the following finite sample estimator of the LHS of eq. (10):

$$\phi_i := \log|\mathcal{D}_O| + \sum_{(s,a)\in\mathcal{D}_O} w_i(s,a)\left[\log w_i(s,a) - \log \frac{c^\star(s)}{1-c^\star(s)}\right],$$

where

$$w_i(s,a) := \text{softmax}_{\mathcal{D}_O}(\delta_i(s,a)) = \frac{\exp\{\delta_i(s,a)\}}{\sum_{(s',a')\in\mathcal{D}_O}\exp\{\delta_i(s',a')\}}.$$

Furthermore, we can optimize the bounded Lagrange multipliers $\sigma(\mu_i)$ by minimizing the loss $\mathcal{L}_\mu := \sum_{i=1}^n \sigma(\mu_i)(\varepsilon - \phi_i)$. Here we use gradient descent to adapt the multipliers $\mu_i$.

### 4.3 Handling Non-Stationary Rewards

To optimize Problem 5 offline, we extend the heuristic by Zahavy et al. (2023) and propose an alternating optimization scheme whose pseudocode is presented in Algorithm 1. While on fixed reward input, the DICE framework computes offline an optimal-dual valued function, in our setting the reward is changing in every iteration. This non-stationarity of reward presents a practical challenge in training the value function and policy. As noted by Vlastelica et al. (2024), the naive approach of training the value function (neural network) to match a moving target tends to be unstable, due to the non-stationary rewards and is further exacerbated by the fact that in each iteration only a single gradient update is made for this reward.

In this work, we address the preceding challenge by conditioning the value function (and policy) on a latent representation of a Functional Reward Encoding (FRE) (Frans et al., 2024), which is pre-trained on random linear functions, random two-layer neural networks with different hidden units, and simple human-engineered rewards. Further details on pre-training are given in Supp. G. In each iteration, given a fixed reward $r$ we compute its encoded FRE latent representation $z_r(S)$ over a subset of state-reward samples $L(r,S) := \{(s, r(s)) : s \in S\}$, where $S$ is subset of states sampled uniformly at random from States$[\mathcal{D}_O]$. Further, to reduce the variance, we sample uniformly at random several state subsets $\{S_1, \ldots, S_m\}$ and take the mean $z_r$ over their FRE latent representations $z_r(S_i)$.

### 4.4 Pseudocode of Dual-Force

In line with standard deep learning practices, the value function and policy are parameterized with neural networks and consequently updated with a single gradient step over batches sampled uniformly at random. Given a batch $\mathcal{B}$, the policy loss becomes $\frac{|\mathcal{D}_O|}{|\mathcal{B}|}\sum_{(s,a)\in\mathcal{B}} w_i^{k+1}(s,a)\log \pi_i(a|s, z_i^k)$.

## 5 Experiments

**Data Collection.** To evaluate our method, we consider the 12 degree-of-freedom quadruped robot SOLO12 (Grimminger et al., 2020) on two robotic tasks in simulation: locomotion and obstacle navigation. For fair comparison and consistency, in terms of quality and diversity of learned skills, our experiments

---

**Algorithm 1** Dual-Force

---

**Input:** state-only expert dataset $\mathcal{D}_E \sim d_E(S)$; offline dataset $\mathcal{D}_O \sim d_O(S, A)$ such that $\mathcal{D}_E \subset \text{States}[\mathcal{D}_O]$; $n$ number of VdW's state-action occupancies; $m$ number of subsets of states; $t$ number of state-reward pairs; Polyak scale $\alpha > 0$
**Initialize:** Sample $w_i^0$ uniformly at random from the probability simplex $\triangle^{|\mathcal{D}_O|}$, for all $i \in \{1, \dots, n\}$

**Pre-train:** a state-discriminator $c^\star : \mathcal{S} \rightarrow (0, 1)$ via optimizing the following objective with the gradient penalty in (Gulrajani et al., 2017) $\max_c \mathbb{E}_{d_E(s)}[\log c(s)] + \mathbb{E}_{d_O(s)}[\log(1 - c(s))]$
**Pre-train:** a Functional Reward Encoding (FRE) $\mathcal{F} : (\mathcal{S} \times \mathcal{R})^m \mapsto \mathcal{Z}$ on state subsets of $\text{States}[\mathcal{D}_O]$ and general unsupervised reward functions as described in Supp. G

**Repeat until convergence:**
  **(Van der Waals Force)**
  **For** each index $i \in \{1, \dots, n\}$:
    compute successor features $\psi_i^k := \sum_{(s,a) \in \mathcal{D}_O} w_i^k(s, a) \phi(s)$
    compute closest distance $\ell_i^k := \|\psi_i^k - \psi_{j_i^\star}^k\|_2$ where $j_i^\star := \arg\min_{j \neq i} \|\psi_i^k - \psi_j^k\|_2$
    compute VdW reward $\beta_i^k(s, a) := (1 - (\ell_i^k / \ell_0)^3) \langle \phi(s), \psi_i^k - \psi_{j_i^\star}^k \rangle$
    compute reward $R_i^k(s, a) := (1 - \sigma(\mu_i^k)) \beta_i^k(s, a) + \sigma(\mu_i^k) \log \frac{c^\star(s)}{1 - c^\star(s)}$
    compute the mean $z_i^k$ over FREs $\{z_i^k(S_j) = \mathcal{F}(L(R_i^k, S_j))\}_{j=1}^m$, where $S_j \sim \text{States}[\mathcal{D}_O]$ with $|S_j| = t$

  **(Value Function and Policy)**
  **For** each index $i \in \{1, \dots, n\}$:
    update with GD the FRE-cond. value function $V_i(\cdot, z_i^k)$ optimizing eq. (14) with the reward $R_i^k$
    compute ratios $w_i(s, a) := \text{softmax}_{\mathcal{D}_O}\left(R_i^k(s, a) + \gamma \mathcal{T} V_i(s, a, z_i^k) - V_i(s, z_i^k)\right)$ for all $s, a \in \mathcal{D}_O$
    compute Polyak average $w_i^{k+1} := (1 - \alpha) w_i^k + \alpha w_i$
    update with GD the FRE-cond. policy $\pi_i(\cdot | \cdot, z_i^k)$ minimizing $\sum_{(s,a) \in \mathcal{D}_O} w_i^{k+1}(s, a) \log \pi_i(a | s, z_i^k)$

  **(Bounded Lagrange Multipliers)**
  **For** each index $i \in \{1, \dots, n\}$:
    compute an estimator $\phi_i := \log |\mathcal{D}_O| + \sum_{(s,a) \in \mathcal{D}_O} w_i^{k+1}(s, a) \left[ \log w_i^{k+1}(s, a) - \log \frac{c^\star(s)}{1 - c^\star(s)} \right]$
  Update with GD $\mu^{k+1}$ minimizing the loss $\sum_{i=1}^n \sigma(\mu_i^k)(\varepsilon - \phi_i)$

---

closely follow the setup in (Vlastelica et al., 2024) and use offline datasets whose collection process is described in Section G "Solo-12 Dataset Collection" of their work. In particular, following (Kim et al., 2022; Ma et al., 2022a; Vlastelica et al., 2024), a state discriminator is learned to differentiate between state demonstrations collected by an expert and from different behavioral policies. To ensure that these behavioral policies provide sufficient diversity while fulfilling a specific task, an online algorithm is run for unsupervised skill discovery subject to value constraints, DOMiNiC (Cheng et al., 2024), and Monte Carlo trajectories are collected from various policy checkpoints throughout the training process. The expert dataset is then mixed into the offline dataset, and the information about which state-action comes from the expert remains hidden to our algorithm. This replicates the experimental setup in (Vlastelica et al., 2024) and allows for a direct comparison between their algorithm (DOI) and ours (Dual-Force).

**Experimental Setup.** For each experiment, we train: a state-discriminator $c^\star$, a Functional Reward Encoding $\mathcal{F}$, a SMODICE-expert (to visualize the target imitation behavior), and diverse skills that satisfy the imitation constraint. To enable recall of all skills learned during training, we store the mean latent reward representation $z_i^k$ at minimal cost for each encountered reward function $R_i^k$, train a conditioned value function and policy on it, and then recall the policy with it during the evaluation process. For both robotic tasks (locomotion and obstacle navigation), the state-discriminator and the SMODICE-expert are trained on the full state space (see Supp. F), forcing the learned skills to imitate the state-only expert demonstrations in their entirety. In contrast, as reported by (Cheng et al., 2024; Vlastelica et al., 2024), it is beneficial in practice to consider a diversity objective with successor features induced by a projection onto the most relevant components of the state space.

**Skills Evaluation.** It is important to note that our problem formulation does not assume access to a reward signal in the offline dataset. However, if the offline dataset contains reward labels, then each skill learned during training can be evaluated off-policy using its corresponding importance ratios $\eta_i$. In this work, we conduct an online evaluation of each learned skill by rolling out 30 Monte Carlo trajectories in simulation. We then compute the mean values of (i) the successor features and (ii) the cumulative return, relative to the reward signal (hidden to our algorithm) used to optimize the expert policy. Afterwards, we report each mean latent reward representation $z_i^k$ that corresponds to a policy $\pi(\cdot|\cdot, z_i^k)$ that achieves at least 50% of the expert's optimal return. While a fraction of the skills $\pi(\cdot|\cdot, z_i^k)$ fail the above criteria, due to intermediate iterations optimizing for diversity, a large fraction of the skills succeed, as the optimization of the imitation constraint takes effect.

**Practical Implementation.** For each state-action occupancy $d_i$ in Problem 5, we train a value function and a policy, parameterized by neural networks. We empirically observed that skill diversity increases and the training procedure stabilizes, when the neural network weights of the value functions (and similarly the policies) are independent across all state-action occupancies. This is efficiently implemented by running the forward pass over all value functions (and policies) in parallel. In the experiments below, we optimize over three state-action occupancies and assign them with the following color map: $d_1$ is orange, $d_2$ is brown, and $d_3$ is red.

**Main Result**. Dual-Force enjoys a strong diversity signal and satisfies the imitation constraint (see Supp. E), recalls all skills encountered during training, and significantly expands the set of diverse skills learned as their number scales with more iterations. In contrast, the DOI algorithm only remembers the last five skills.

## 5.1 Locomotion Task

**Data Collection.** The expert dataset is collected from a uni-modal expert trained to walk straight with constant linear velocity and middle base-height. The offline dataset contains non-expert behaviors achieving constant linear and angular velocity, and walking movements with different base-heights (low, middle, orange).

**Diversity Objective.** The state space is projected onto the joint position components represented by a 12-dimensional vector, which captures the movements of each of the four legs: front left, front right, hind left, and hind right - and includes hip abduction/adduction, hip flexion/extension, and knee flexion/extension. The joint positions serve as a proxy for body height, which is missing from the offline dataset.

**Results [Uni-Modal].** Figure 2 demonstrates that our method finds diverse skills that achieve constant linear and angular velocity and, more importantly, recovers all base-height movements in the offline dataset, including the expert behavior. Figure 3 (a,b) shows that the successor features of the learned skills are clustered into three groups (according to the base-height). While the $\ell_2$ pairwise distance between the successor features within a cluster is small, the distance between clusters is large. Here we use the UMAP (McInnes et al., 2018) algorithm to project the successor features into 2D space.

## 5.2 Obstacle Navigation Task

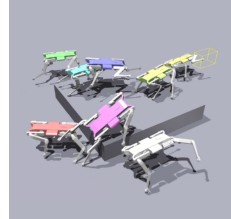

**Data Collection.** The expert dataset is collected from a multi-modal expert that initialized in front of a box is trained to reach a target position behind the box by either going over the box or surrounding it from the left or the right side. The offline dataset contains various non-expert behaviors collected at different time checkpoints during the expert's training procedure. It is important to note that these behaviors may not reach the target position, nor do they have to remain standing for the entire episode.

**Diversity Objective.** The state space is projected onto the "base linear velocity" and "base angular velocity" components, each represented by a 3-dimensional vector. This choice encourages diversity, as most trajectories in the offline dataset have similar body heights on the ground, but approach the obstacle from different directions and at different speeds.

**Results [Multi-Modal].** In Figure 3 (c,d) we show that the successor features of the learned skills are well separated. Figure 4 demonstrates that our method finds diverse skills that reach the target position. Furthermore, the learned skill set captures all expert behaviors and various modalities of the offline dataset.

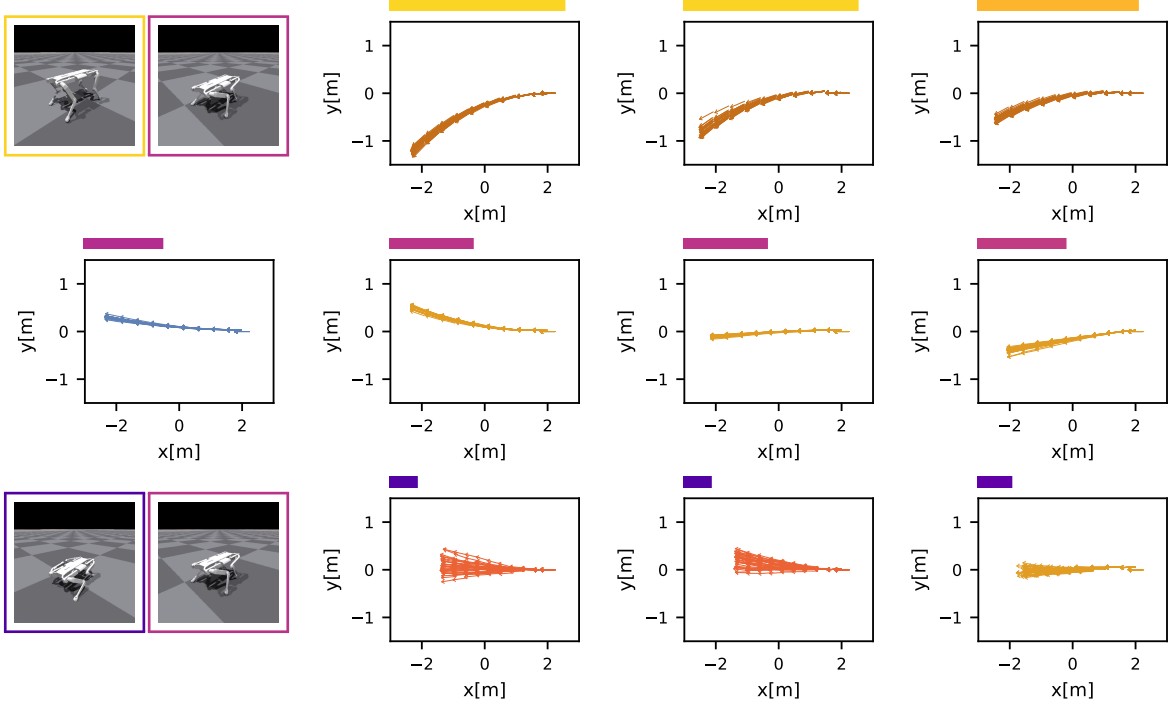

Figure 2: A performance benchmark of skills learned in the locomotion task. The SMODICE-expert walks with constant base and angular velocity, and with middle base-height. The learned skills recovers all base-height movements (low, middle, high) and have different angular velocity. The colored horizontal bar at the top of each skill plot indicates the corresponding average base-height.

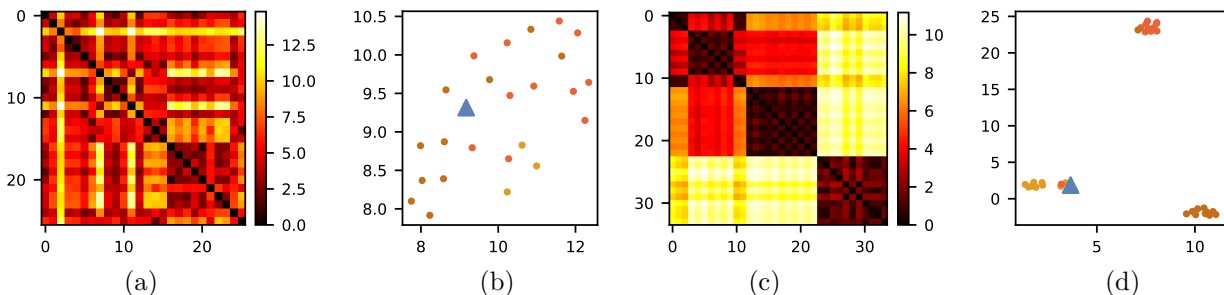

Figure 3: Tasks: (a,b) Locomotion and (c,d) Navigation. The blue triangle is the SMODICE-expert and the colored dots are the learned skills. (a,c) Successor features pair-wise $\ell_2$ distances between skills. The first row (column) is SMODICE-expert, and all other rows (columns) are learned skills. (b,d) UMAP projection of successor features into 2D space.

Then in Figure 5 we show that it also provides a robust solution for reaching the target position even in an adversarial setting where additional fence obstacles partially block the path to the target position.

## 6 Related Work

**Skill discovery.** In the unconstrained and online setting, various approaches of unsupervised skill discovery algorithms have been proposed (Eysenbach et al., 2019; Campos et al., 2020; Achiam et al., 2018; Strouse et al., 2022). These methods struggle to learn large numbers of skills (Campos et al., 2020; Achiam et al., 2018). Sharma et al. (2020) make use of skill predictability as a proxy for guiding skill discovery. Strouse et al. (2022) introduce an ensemble-based information gain formulation. Kim et al. (2021) are able to learn disentangled and interpretable skill representations. These methods are inherently online and require extensive interaction with the environment. Although there are offline methods based on skill predictability (Liu et al., 2023), they either require expert demonstrations with actions for imitation, or use clustering techniques to extract skills already present in the unlabeled dataset, thus failing to generate novel skills.

**Unsupervised RL.** The goal of unsupervised RL is to allow agents to explore and learn meaningful representations and skills in an environment without explicit rewards or supervision. In their seminal work, Dayan (1993); Barreto et al. (2017) proposed "successor features", a value function representation that decouples the dynamics of the environment from the rewards, bringing transfer in RL across tasks. These representations have also been combined with intrinsic motivation to enhance diversity (Gregor et al., 2017; Barreto et al., 2017; Hansen et al., 2020).

**Quality-Diversity.** Discovering a diverse set of optimal or near-optimal policies can improve the robustness and generalization of RL agents. The literature largely encompasses two main algorithmic families: MAP-Elites (Cully et al., 2015; Mouret and Clune, 2015) and novelty search with local competition (Lehman and Stanley, 2011). These algorithms use evolutionary strategies to maintain and adapt a collection of policies, aiming to balance the quality and diversity trade-off (Cully, 2019; Pugh et al., 2016; Tarapore et al., 2016).

**Constrained skill discovery.** We are not the first to consider a constrained diversity maximization approach. Zahavy et al. (2023) proposed an online skill discovery method that handles value constraints. Cheng et al. (2024) extended their approach to the setting of multiple constraints, while still remaining in an online setting. The diversity objectives in (Zahavy et al., 2023; Cheng et al., 2024) both use the VdW force. Vlastelica et al. (2024) proposed an offline algorithm for maximizing a mutual information objective subject to imitation constraints.

**Off-policy estimation.** Our work builds upon the "DIstribution Correction Estimation (DICE)" framework and provides a robust importance sampling technique for off-policy learning (Nachum and Dai, 2020) which finds applications in computing policy gradients from off-policy data (Nachum et al., 2019), offline imitation learning with imperfect demonstrations (Kim et al., 2022; Ma et al., 2022a), and off-policy evaluation (Dai et al., 2020). Our off-policy approach is also similar to (Lee et al., 2021; 2022; Vlastelica et al., 2024).

## 7 Conclusion

In this work, we introduce Dual-Force, a novel offline algorithm designed to maximize diversity under imitation constraints based on expert state demonstrations. Our main contribution is to propose off-policy estimator of the Van der Waals force, eliminating the need for a skill discriminator, thus enhancing training stability and efficiency. Furthermore, by conditioning the value function and policy on a pre-trained Functional Reward Encoding, our method handles non-stationary rewards better and provides zero-shot recall of all skills encountered during training. Experimental results demonstrate the effectiveness of the proposed method in generating diverse skills for robotic tasks in simulation. Notably, the learned skill set includes various modalities derived from both expert and offline datasets, highlighting the method's robust capabilities for versatile skill discovery.

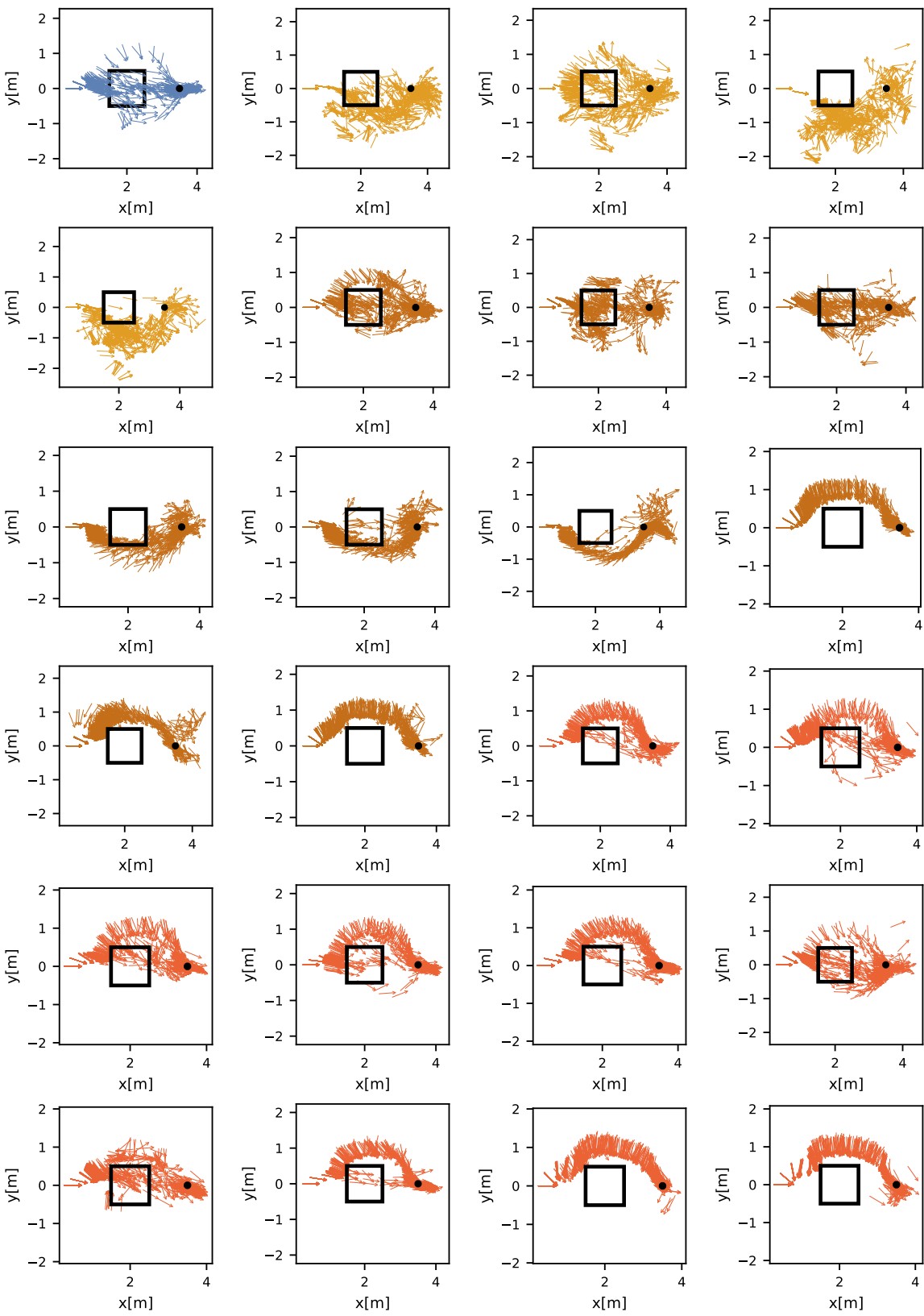

Figure 4: A performance benchmark of skills learned in the obstacle navigation task, where the SMODICE-expert is initialized sin front of a box of height 0.2m and reaches a target position behind the box. The learned skills exhibit diverse behaviors that cover various modalities of expert and offline datasets.

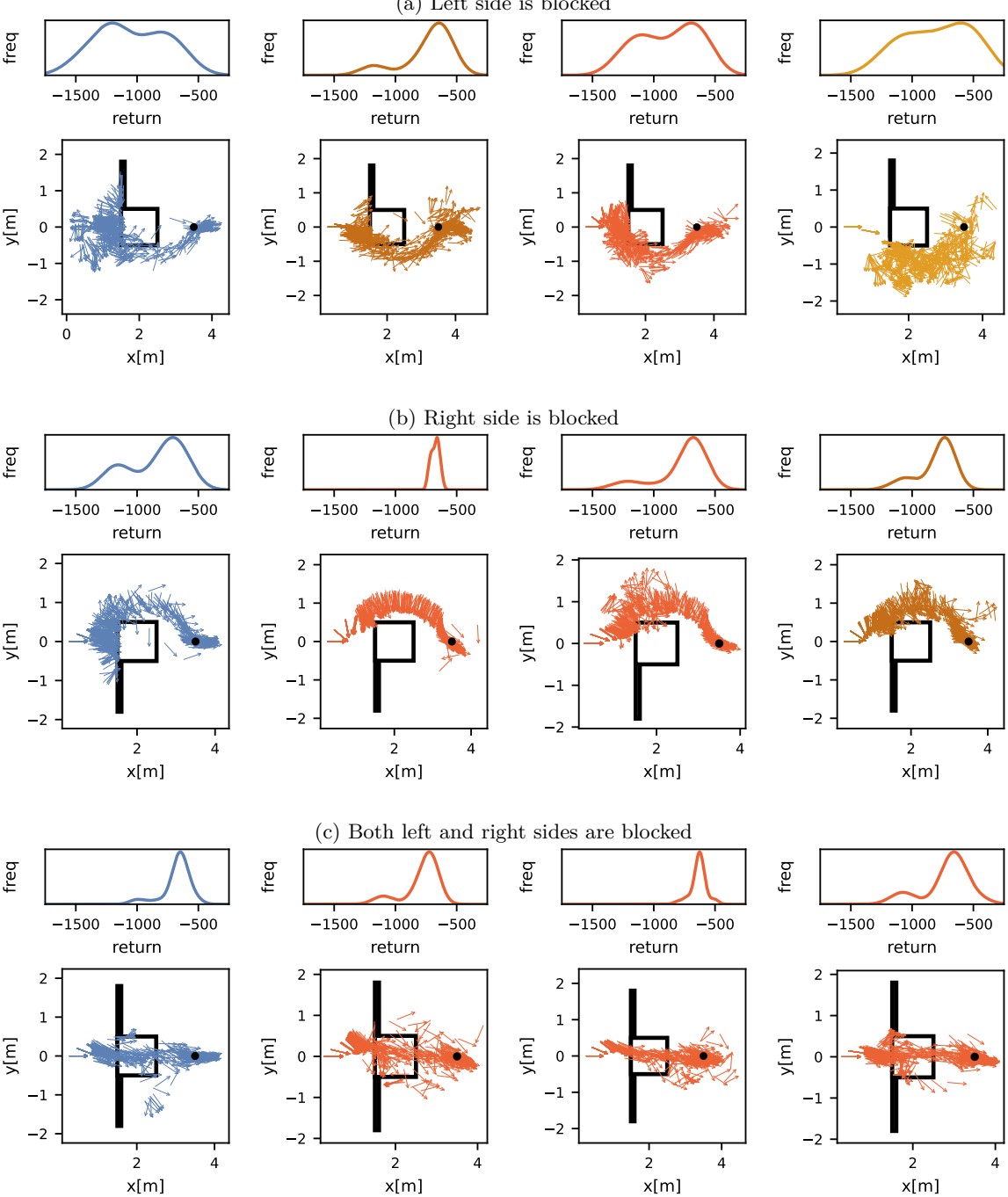

Figure 5: A performance benchmark with additional fence obstacles, of height 0.6m, partially blocking the path from (a) the left side, (b) the right side, or (c) both the left and right sides. Among the diverse skills learned, there are several that outperform the SMODICE-expert in (a,b) and perform on par with the SMODICE-expert in (c).

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

# Supplementary for Dual-Force: Enhanced Offline Diversity Maximization under Imitation Constraints

## A Imitation Constraint Relaxation

Our analysis make use of the following assumption.

**Assumption A.1** (Expert coverage). *We assume that $d_E(s) > 0$ implies $d_O(s) > 0$.*

**Lemma A.2** (State-only KL Estimator). *Under Theorem A.1, we have*

$$D_{\mathrm{KL}}\left(d_i(S)||d_E(S)\right) \leq -\mathbb{E}_{d_i(s)}\left[\log \frac{d_E(s)}{d_O(s)}\right] + D_{\mathrm{KL}}\left(d_i(S,A)||d_O(S,A)\right) \tag{S1}$$

*Proof.* The statement follows by combining Claim A.4 and A.5. $\square$

**Corollary A.3** (Structural). *Under Theorem A.1, the RHS of equation S1 is estimated by*

$$\mathbb{E}_{d_O(s,a)}\left[\eta_i(s,a)\left(\log \eta_i(s,a) - \log \frac{c^\star(s)}{1 - c^\star(s)}\right)\right].$$

*Proof.* The statement follows by combining Lemma A.2 and Claim A.6. $\square$

### A.1 Useful Facts

**Claim A.4.** *It holds that*

$$D_{\mathrm{KL}}\left(d_{\pi_1}(S,A)||d_{\pi_2}(S,A)\right) = D_{\mathrm{KL}}\left(d_{\pi_1}(S)||d_{\pi_2}(S)\right) + \mathbb{E}_{d_{\pi_1}(s)}D_{\mathrm{KL}}\left(\pi_1(\cdot|s)||\pi_2(\cdot|s)\right)$$

*Proof.* We have

$$
\begin{aligned}
D_{\mathrm{KL}}\left(d_{\pi_1}(S,A)||d_{\pi_2}(S,A)\right) &= \mathbb{E}_{d_{\pi_1}(s,a)}\left[\log \frac{d_{\pi_1}(s,a)}{d_{\pi_2}(s,a)}\right] = \mathbb{E}_{d_{\pi_1}(s,a)}\left[\log \frac{d_{\pi_1}(s)\pi_1(a|s)}{d_{\pi_2}(s)\pi_2(a|s)}\right] \\
&= \mathbb{E}_{d_{\pi_1}(s,a)}\left[\log \frac{d_{\pi_1}(s)}{d_{\pi_2}(s)}\right] + \mathbb{E}_{d_{\pi_1}(s)}\mathbb{E}_{\pi_1(a|s)}\left[\log \frac{\pi_1(a|s)}{\pi_2(a|s)}\right] \\
&= D_{\mathrm{KL}}\left(d_{\pi_1}(S)||d_{\pi_2}(S)\right) + \mathbb{E}_{d_{\pi_1}(s)}D_{\mathrm{KL}}\left(\pi_1(\cdot|s)||\pi_2(\cdot|s)\right)
\end{aligned}
$$

$\square$

**Claim A.5.** *Under Theorem A.1, we have*

$$D_{\mathrm{KL}}\left(d_i(S)||d_E(S)\right) = -\mathbb{E}_{d_i(s)}\left[\log \frac{d_E(s)}{d_O(s)}\right] + D_{\mathrm{KL}}\left(d_i(S)||d_O(S)\right)$$

*Proof.* We have

$$
\begin{aligned}
D_{\mathrm{KL}}\left(d_i(S)||d_E(S)\right) &= \mathbb{E}_{d_i(s)}\left[\log \frac{d_i(s)}{d_E(s)}\right] \\
&= \mathbb{E}_{d_i(s)}\left[\log \frac{d_i(s)}{d_O(s)} \cdot \frac{d_O(s)}{d_E(s)}\right] \\
&= -\mathbb{E}_{d_i(s)}\left[\log \frac{d_E(s)}{d_O(s)}\right] + D_{\mathrm{KL}}\left(d_i(S)||d_O(S)\right)
\end{aligned}
$$

$\square$

**Claim A.6.** *Let $\eta_i(s,a) = \frac{d_i(s,a)}{d_O(s,a)}$ for all $(s,a) \in \mathcal{D}_O$, and $c^\star(s) = \frac{d_E(s)}{d_E(s)+d_O(s)}$ for all $s \in \mathcal{D}_E \cup \mathcal{D}_O$*

$$\mathbb{E}_{d_i(s)}\left[\log \frac{d_E(s)}{d_O(s)}\right] \approx \mathbb{E}_{d_O(s,a)}\left[\eta_i(s,a)\log\frac{c^\star(s)}{1-c^\star(s)}\right]$$

*Proof.* We have

$$
\begin{aligned}
\mathbb{E}_{d_i(s)}\left[\log\frac{d_E(s)}{d_O(s)}\right] &= \mathbb{E}_{d_i(s)}\mathbb{E}_{\pi(a|s)}\left[\log\frac{d_E(s)}{d_O(s)}\right] = \mathbb{E}_{d_i(s,a)}\left[\log\frac{d_E(s)}{d_O(s)}\right]\\
&\approx \mathbb{E}_{d_O(s,a)}\left[\eta_i(s,a)\log\frac{c^\star(s)}{1-c^\star(s)}\right]
\end{aligned}
$$

$\square$

## B  KL Estimator

Recall that the weight $w_i(s,a)$ is defined w.r.t. a fixed dataset $\mathcal{D}_0$ and reads

$$w_i(s,a) = \mathrm{softmax}_{\mathcal{D}_O}(\delta_i(s,a)) = \frac{\exp\{\delta_i(s,a)\}}{\sum_{(s',a')\in\mathcal{D}_O}\exp\{\delta_i(s',a')\}},$$

where the TD error $\delta_i(s,a) = R_i^\mu(s,a) + \gamma\mathcal{T}V_i^\star(s,a) - V_i^\star(s)$. In contrast, the importance ratio $\eta_i(s,a)$ is defined in terms of the expectation of the state-action occupancy $d_O$, namely

$$\eta_i(s,a) = \mathrm{softmax}_{d_O(s,a)}(\delta_i(s,a)) = \frac{\exp\{\delta_i(s,a)\}}{\mathbb{E}_{d_O(s',a')}\exp\{\delta_i(s',a')\}}.$$

**Claim B.1.** *Given an offline dataset $\mathcal{D}_O$ sampled u.a.r. from state-action occupancy $d_O$, an estimator of the importance ratio $\eta_i(s,a)$ is given by $\widetilde{\eta}_i(s,a) := |\mathcal{D}_O|w_i(s,a)$.*

*Proof.* Combining $\frac{1}{|\mathcal{D}_O|}\sum_{(s',a')\in\mathcal{D}_O}\exp\{\delta_i(s',a')\}$ is an estimator of the expectation $\mathbb{E}_{d_O(s',a')}\exp\{\delta_i(s',a')\}$ and the definition of weight $w_i(s,a)$ we have

$$
\begin{aligned}
\eta_i(s,a) &= \frac{\exp\{\delta_i(s,a)\}}{\mathbb{E}_{d_O(s',a')}\exp\{\delta_i(s',a')\}}\\
&\approx \frac{\exp\{\delta_i(s,a)\}}{\frac{1}{|\mathcal{D}_O|}\sum_{(s',a')\in\mathcal{D}_O}\exp\{\delta_i(s',a')\}} = |\mathcal{D}_O|w_i(s,a) = \widetilde{\eta}_i(s,a).
\end{aligned}
$$

$\square$

**Lemma B.2.** *The KL-divergence $D_{\mathrm{KL}}(d_i(S,A)||d_O(S,A))$ admits the following estimator,*

$$\log|\mathcal{D}_O| + \sum_{(s,a)\in\mathcal{D}_O} w_i(s,a)\log w_i(s,a).$$

*Proof.* Combining the definition of $\eta_i(s,a) = d_i(s,a)/d_O(s,a)$ and Theorem B.1, we have

$$
\begin{aligned}
D_{\mathrm{KL}}\left(d_i(S,A)||d_O(S,A)\right) &= \mathbb{E}_{d_i(s,a)}\log\eta_i(s,a)\\
&= \mathbb{E}_{d_O(s,a)}\eta_i(s,a)\log\eta_i(s,a)\\
&\approx \frac{1}{|\mathcal{D}_O|}\sum_{(s,a)\in\mathcal{D}_O}\widetilde{\eta}_i(s,a)\log\widetilde{\eta}_i(s,a)\\
&= \sum_{(s,a)\in\mathcal{D}_O}w_i(s,a)\log\left(|\mathcal{D}_O|w_i(s,a)\right)\\
&= \log|\mathcal{D}_O| + \sum_{(s,a)\in\mathcal{D}_O}w_i(s,a)\log w_i(s,a)
\end{aligned}
$$

$\square$

## C   Successor Features as Diversity Measure

**Lemma C.1** (Convex Diversity Objective). *Let $\Phi \in \mathbb{R}^{d \times (S \times A)}$ be a feature map and $d_i \in \triangle^{S \times A}$ be a probability distribution. Then for the feature vector $\psi_i = \Phi d_i \in \mathbb{R}^d$ we have*

$$\nabla_{d_i} \frac{1}{2} \|\psi_i - \psi_j\|_2^2 = \Phi^T (\psi_i - \psi_j).$$

*Further, the corresponding Hessian is positive semi-definite matrix, i.e.,*

$$\nabla_{d_i} \Phi^T \Phi (d_i - d_j) = \Phi^T \Phi \succeq 0.$$

*In particular, $\frac{1}{2}\|\Phi d_i - \Phi d_j\|_2^2$ is a convex function w.r.t. $d_i$.*

*Proof.* Observe that

$$
\begin{aligned}
\nabla_{d_i(s,a)} \frac{1}{2} \sum_{\ell=1}^{n} (\Phi_{\ell,:} d_i - \Phi_{\ell,:} d_{\pi_2})^2 &= \sum_{\ell=1}^{n} (\Phi_{\ell,:} d_i - \Phi_{\ell,:} d_j)[\phi(s,a)]_{\ell} \\
&= \left( \sum_{\ell=1}^{n} \Phi_{\ell,:} [\phi(s,a)]_{\ell} \right)(d_i - d_j) \\
&= \phi(s,a)^T \Phi (d_i - d_j) \\
&= \phi(s,a)^T (\psi_i - \psi_j)
\end{aligned}
$$

Hence, we have

$$
\begin{aligned}
\nabla_{d_i} \frac{1}{2} \|\psi_i - \psi_j\|_2^2 &= \nabla_{d_i} \frac{1}{2} \|\Phi d_i - \Phi d_j\|_2^2 \\
&= \nabla_{d_i} \frac{1}{2} \sum_{\ell=1}^{n} (\Phi_{\ell,:} d_i - \Phi_{\ell,:} d_j)^2 \\
&= \sum_{\ell=1}^{n} \Phi_{\ell,:} (d_i - d_j) \Phi_{\ell,:}^T \\
&= \left[ \sum_{\ell=1}^{n} \Phi_{\ell,:}^T \Phi_{\ell,:} \right](d_i - d_j) \\
&= \Phi^T \Phi (d_i - d_j) \\
&= \Phi^T (d_i - d_j)
\end{aligned}
$$

and

$$\nabla_{d_i} \Phi^T \Phi (d_i - d_j) = \Phi^T \Phi.$$

$\square$

## D   Reproducibility

For the implementation of Dual-Force we used the PyTorch Autograd framework. The offline datasets were collected by Vlastelica et al. (2024) and we used Isaac Gym to evaluate the learned skills. The training was performed on an NVIDIA GeForce RTX 2080 Ti graphics card, and computations took in real-time:

- FRE transformer (see Supp. G): Locom (10h, batch 2048), Obstacle-Navi (10h, batch 1280)
- Skill-discriminator + SMODICE-expert: Locom (0.3h, batch 8192), Obstacle-Navi (1h, batch 8192)
- Dual-Force – Locom (0.5h, batch 8192), Obstacle-Navi (1h, batch 2560)

The SOLO12 robot is developed as part of the Open Dynamic Robot Initiative Grimminger et al. (2020).

# E   Training Metrics

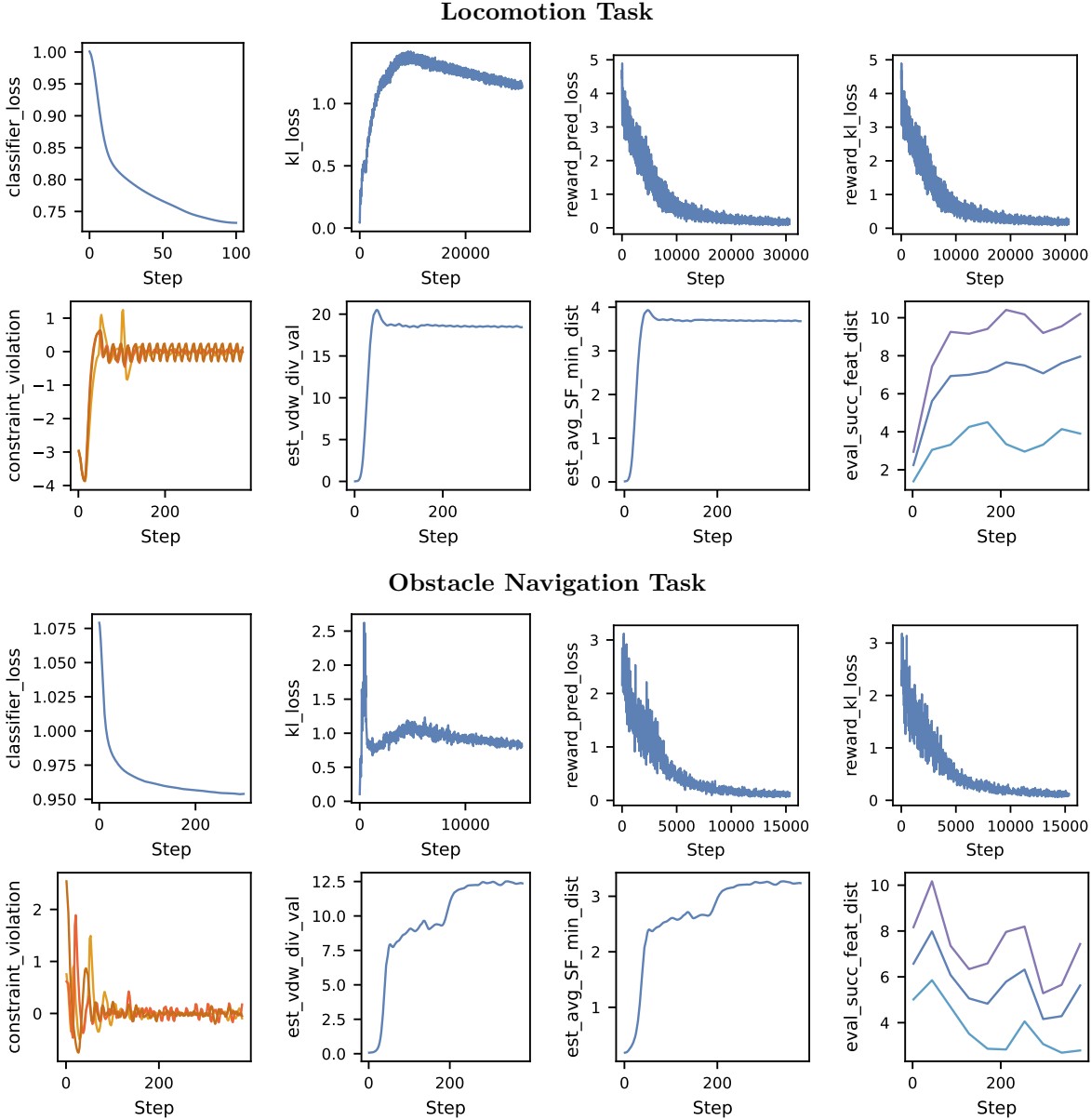

Figure S1: Training metrics of State Discriminator, Functional Reward Encoding, and Dual-Force algorithm.

In Figure S1, we report important metrics computed during the training of the state-discriminator $c^\star$, the Functional Reward Encoding $\mathcal{F}$, and the Dual-Force algorithm. In particular, the four subplots in the first row show: (1) a training loss of the state-discriminator $c^\star$; and (2,3,4) training losses of the Functional Reward Encoding. The subplots in the second row show four evaluation metrics of the Dual-Force algorithm: (5) an estimated constraint violations of the three state-action occupancies $d_1$, $d_2$, and $d_3$; (6,7) Van der Waals diversity measures – an estimated VdW objective value (see eq. (9)) and an estimated average of minimum $\ell_2^2$ distance between successor features of distinct $d_1$, $d_2$, and $d_3$ (see eq. (8)); and (8) an online evaluation of {minimum, average, maximum} successor features $\ell_2^2$ distance among $d_1$, $d_2$, and $d_3$.

In the VdW diversity objective, the state space is projected onto different components depending on the task at hand (locomotion or obstacle navigation). For the locomotion task, the state space is projected onto the

joint position components, represented by a 12-dimensional vector (see Supp. F), which serves as a proxy for body height, as this information is missing from the offline dataset. For the obstacle navigation task, the state space is projected onto the "base linear velocity" and "base angular velocity" components, each represented by a 3-dimensional vector, as most trajectories in the offline dataset have similar body heights on the ground, but approach the obstacle from different directions and at different speeds.

**Locomotion.** The offline dataset is composed of 248751 expert and 996000 behavior transitions, in total 1244751. The state space has 48-dimensions, the VdW diversity objective projects the state space onto the "joint positions" (12-dimensional vector), the target minimum distance between SFs of learned skills is $\ell_0 = 6.0$, and the target imitation constraint threshold is $\epsilon = 4.0$.

**Obstacle Navigation.** The offline dataset is composed of 654501 expert and 749501 behavior transitions, in total 1404002. The state space has 171-dimensions, the VdW diversity objective projects the state space onto the "base linear velocity" and "base angular velocity" (each with 3-dimensions), the target minimum distance between SFs of learned skills is $\ell_0 = 4.0$, and target imitation constraint threshold is $\epsilon = 1.0$.

## F  Solo12 State Space

For fair comparison and consistency, in terms of quality and diversity of learned skills, our experiments closely follow the setup in (Vlastelica et al., 2024) and use offline datasets whose collection process is described in Section G "Solo-12 Dataset Collection" of their work.

**State Space.** For the locomotion task, the state space has 48 dimensions:

[3-dims each] "base linear velocity", "base angular velocity", "projected gravity", "commanded velocity";

[12-dims each] "joint positions", "joint velocity", "previous action".

The "joint positions" components capture the movements of each of the four legs: front left, front right, hind left, and hind right - and includes hip abduction/adduction, hip flexion/extension, and knee flexion/extension. The "joint velocity" components have similar semantics to the "joint positions". The "previous action" encodes the target joint positions from the previous time step.

For the obstacle navigation task, the state space contains 171 dimensions:

- the above locomotion state with 48 dimensions;
- "surrounding height map" of the robot with 121 dimensions;
- "remaining time" until the end of the episode with 1 dimension.

## G  Pre-training of Functional Reward Embedding

We pretrain the FRE model following the approach of Frans et al. (2024). For both the locomotion task and the obstacle navigation task, to ensure wide diversity of general unsupervised reward functions, we generate a list of rewards as follows:

30x linear functions with random weights

30x two-layered perceptron (MLP) neural networks with random weights and hidden units in $[(128, 64), (128, 128), (256, 128), (256, 256), (512, 256), (512, 512)]$

27x combination of simple human-engineered rewards that incentivize constant base and angular velocity in different directions, and different joint angle heights.

It is important to note that the above FRE latent representation *cannot* affect the diversity of skills learned by Algorithm 1, but rather serves only as a hash map that assigns a unique label to each reward so that the FRE-conditional value function and policy can better handle the training of the non-stationary rewards.

