# OpenReview forum: "Dual-Force: Enhanced Offline Diversity Maximization under Imitation Constraints"
_TMLR — Rejected by TMLR_

### Review · Reviewer_PR4G · 2025-02-07

**Summary Of Contributions:**

This paper addresses the problem of learning diverse behaviors in online imitation learning. In the proposed method, a latent-conditioned policy is trained to model multiple behaviors. In the proposed method, the diversity of behaviors is encouraged by using repulsive forces based on the Van der Waals (VdW) force. To address the distribution shift problem in offline imitation learning, the proposed method employs the DIstribution Correction Estimation (DICE) framework. In addition, the latent space is learned using Functional Reward Encoding (FRE). The proposed method was evaluated on locomotion and navigation tasks.

**Audience:**

No

**Broader Impact Concerns:**

I do not have any concerns about the ethical implications.

**Claims And Evidence:**

No

**Requested Changes:**

- I believe it is essential to include a comparison with baseline methods. For example, CLUE, proposed in [R1], could serve as a strong baseline for learning diverse behaviors in offline imitation learning. Additionally, training a latent-conditioned policy with SMODice and incorporating the Van der Waals (VdW) force, as proposed by Zahavy et al. (2023), could also provide a valuable baseline.

[R1] CLUE: Calibrated Latent Guidance for Offline Reinforcement Learning

- The proposed method consists of multiple components, but their individual contributions remain unclear. Conducting an ablation study is crucial to evaluate the effect of each component in the proposed method.

- While I understand that quantifying the diversity of behaviors is challenging, please consider exploring a method to measure the diversity of the learned skills and compare it with baseline methods.

- Section 4.2 appears to be based on (Nachum and Dai, 2020; Ma et al., 2022a; 2022b). I believe it is necessary to explicitly cite these papers within Section 4.2 as well.

**Strengths And Weaknesses:**

Strengths:
- The proposed method appears solid and well-designed.

Weaknesses:
- The novelty of the proposed method is unclear.
- There is no quantitative evaluation of the proposed method.
- There is no comparison with baseline methods.
- The presentation lacks clarity.

The paper appears to be closely related to Vlastelica et al. (2024). Although the reference section cites a publication in the Reinforcement Learning Conference 2024, I was unable to find "Diverse Offline Imitation Learning" in the publication list of RLC ( https://rlj.cs.umass.edu/2024/2024issue.html ).
Since it is unclear which parts have already been published, I am unable to accurately assess the contribution of this paper.

While I appreciate the authors' efforts in developing the proposed framework, the method itself seems somewhat incremental. Although I understand that it enables the learning of diverse behaviors in offline imitation learning, the specific advantages of the proposed approach are not clearly evident.

---

> ### Author Response · Authors · 2025-02-17
> **Response to Reviewer PR4G - Weakness**
>
> Thank you for your valuable suggestions and constructive feedback! We answer the raised questions below.
>
> > C1. The novelty of the proposed method is unclear.
>
> As discussed in the last paragraph of the Introduction section, a novel offline method for diversity maximization under imitation constraints is proposed. Notably, it has the following three beneficial properties:
> 1. It addresses a challenge posed by Vlastelica et al. (2024), which states that training a skill discriminator in the offline setting is difficult, results in a vanishing diversity signal, and leads to unstable training. In contrast, the Dual-Force algorithm does not train a skill-discriminator and provides a strong diversity signal throughout the whole execution, based on a physically inspired VdW force.
> 2. The proposed method extends an online framework for diversity maximization under linear constraints of Zahavy et al. (2023). In particular, we consider non-linear (KL divergence) imitation constraints and combine the DICE framework of Nachum and Dai (2020) to design an offline algorithm.
> 3. It addresses a limitation in the methods of Zahavy et al. (2023) and Vlastelica et al. (2024), which require a priori to fix the number of different skills to be learned, e.g. five. While running these algorithms, the learned policy only captures the last five skills (since there are five fixed placeholders) and forgets everything else. Moreover, the runtime complexity depends linearly on the number of skills. In contrast, Dual-Force learns and provides zero-shot recall of all skills encountered during training! This significantly expands the set of diverse skills learned. It is crucial to emphasize that the proposed algorithm is independent of the "number of skills" parameter, i.e., the number of learned skills scales with the number of iterations one can afford to execute. The Dual-Force algorithm achieves this by conditioning the value function and policy on a pre-trained "Functional Reward Encoding" by Frans et al. (2024), which maps each skill reward to a compact latent representation in each phase and, as a byproduct, allows for better handling of nonstationary rewards.
> We will update the introduction to reflect this discussion and provide empirical evidence of a strong diversity signal throughout the execution.
>
> > C2. The paper appears to be closely related to Vlastelica et al. (2024). Although the reference section cites a publication in the Reinforcement Learning Conference 2024, I was unable to find "Diverse Offline Imitation Learning" in the publication list of RLC ( https://rlj.cs.umass.edu/2024/2024issue.html ). Since it is unclear which parts have already been published, I am unable to accurately assess the contribution of this paper.
>
> Thank you for your important comment! We regret that there was a typo in the reference section, this caused unnecessary confusion. The correct link is https://rlj.cs.umass.edu/2024/papers/RLJ_RLC_2024_169.pdf
>
> > C3. There is no quantitative evaluation of the proposed method.
>
> We respectfully disagree. As discussed in the Experiments section, the following quantitative evaluations were performed:
> - The heat maps in Fig. 3 show the pairwise L2 distances between the successor features (SFs) of the learned skills for both the Locomotion and Obstacle Navigation tasks. It is crucial to emphasize that this is a well-established quantitative measure of skill distinctiveness. In addition, we project the SFs of the learned skills into 2D space using the UMAP algorithm.
> - In Fig. 2 (Locomotion) and Fig. 4 (Obstacle Navigation), we rollout 30 Monte Carlo trajectories in simulation for each learned skill and plot the observed behaviors. For the Locomotion task, the learned skills recover all base-height movements (low, middle, high) and have different angular velocity. For the obstacle navigation task, the learned skills exhibit diverse behaviors covering different modalities of the multimodal expert and offline datasets. Furthermore, a comparison with the SMODICE-expert of Ma et al. (2022) is provided, which only optimizes the expert imitation.
> - In Fig. 5, it is shown that among the diverse skills learned, there are several robust to environment perturbation (adding a blocking fence) that either outperform or are on par with the SMODICE-expert, demonstrating the effectiveness of the proposed method.
>
> > C4. There is no comparison with baseline methods.
>
> Please see our response to requests R1, R2, R3.
>
> > C5. The presentation lacks clarity.
>
> We would like to ask the reviewer to be more specific so that we can address this concern.
>
> > C6. While I appreciate the authors' efforts in developing the proposed framework, the method itself seems somewhat incremental. Although I understand that it enables the learning of diverse behaviors in offline imitation learning, the specific advantages of the proposed approach are not clearly evident.
>
> We hope our response to comment C1 (and R3) clarifies the three specific advantages.

---

> > ### Author Response · Authors · 2025-02-17
> > **Response to Reviewer PR4G - Requested Changes**
> >
> > > R1. I believe it is essential to include a comparison with baseline methods. For example, CLUE, proposed in [R1], could serve as a strong baseline for learning diverse behaviors in offline imitation learning. // [R1] CLUE: Calibrated Latent Guidance for Offline Reinforcement Learning
> >
> > This is an excellent point. However, our experiments closely follow the setup in Vlastelica et al. (2024), see Fig. S4 (locomotion) and Fig. 6 (obstacle navigation) in their paper, and thus serve as a valid comparison to previous work. Moreover, the presented experimental results in Figures 2,3,4 and 5 clearly support our main statement that the Dual-Force algorithm learns a significantly larger set of diverse skills and can zero-shot recall them!
> >
> > Thank you for pointing out the work of Liu et al. (2023), we will discuss it in the paper. However, it is important to note that there are two major differences in the problem formulations studied in CLUE that make the direct comparison with our work inapplicable.
> > 1. “The expert dataset contains actions.” In contrast, our setting poses an additional challenge by considering imitation from state-only demonstrations, i.e., the Dual-Force algorithm does not have access to labeled expert actions.
> > 2. “In the unsupervised offline RL setting, it clusters the data into multiple classes.” In contrast, the Dual-Force algorithm does not attempt to cluster the offline data, but rather to find diverse pairwise distinct skills (measured by successor features), which may lead to learning behaviors that were not observed in the offline dataset. For example, as shown in Figures 2 and 4, the agent learns to combine movements with different angular velocities.
> >
> > > R2. Additionally, training a latent-conditioned policy with SMODice and incorporating the Van der Waals (VdW) force, as proposed by Zahavy et al. (2023), could also provide a valuable baseline.
> >
> > We believe this is a misunderstanding. The Dual-Force algorithm is an offline extension of the DOMINO algorithm proposed by Zahavy et al. (2023) with a VdW diversity objective and a KL-divergence constraint (instead of a linear Value constraint). Furthermore, this imitation constraint can be seen as a relaxation of the SMODICE minimization objective of Ma et al. (2022). We claim that our algorithm is a principled way to combine DOMINO and SMODICE.
> >
> > > R3. The proposed method consists of multiple components, but their individual contributions remain unclear. Conducting an ablation study is crucial to evaluate the effect of each component in the proposed method.
> >
> > Thank you for raising this important issue. However, as explained in detail in our response to comment C1, the design choices of the Dual-Force algorithm make the ablation study self-evident. In particular:
> > 1. The VdW diversity objective removes the necessity to train a skill discriminator, addressing a limitation posed by Vlastelica et al. (2024);
> > 2. A detailed explanation of how to estimate the VdW diversity objective offline is given in Section 4, and a formal proof of how to compute an offline estimator of the KL-divergence term in the imitation constraint is given in Appendix B.
> > 3. It is crucial to emphasize that the pre-trained FRE mapping serves as a form of hashmap, which allows us to learn and zero-shot recall all skills encountered during training! In sharp contrast, previous work can only recall the currently learned skills at the last iteration, i.e., five.
> >
> > Following the reviewer's request, we will provide empirical evidence in the Appendix for both locomotion and obstacle navigation tasks to support the claims that:
> > - the diversity signal remains strong throughout the execution of the algorithm; and
> > - the constraint violations fluctuate close to the specified target threshold $\epsilon$.
> >
> > > R4. While I understand that quantifying the diversity of behaviors is challenging, please consider exploring a method to measure the diversity of the learned skills and compare it with baseline methods.
> >
> > Thank you for your constructive feedback. It is important to emphasize that the reported in Fig. 3 pairwise L2 distance between the successor features of the learned skills is a well-established quantitative measure of skill distinctiveness. Furthermore, the comparison of our results with the previous work of Vlastelica et al. (2024) serves as a valid comparison. Notably, our algorithm learns and zero-shot recalls significantly more diverse skills than previous work.
> >
> > > R5. Section 4.2 appears to be based on (Nachum and Dai, 2020; Ma et al., 2022a; 2022b). I believe it is necessary to explicitly cite these papers within Section 4.2 as well.
> >
> > Following the reviewer's request, we will update the manuscript with appropriate citations.

---

> > > ### Comment · Reviewer_PR4G · 2025-03-03
> > >
> > > > We believe this is a misunderstanding. The Dual-Force algorithm is an offline extension of the DOMINO algorithm proposed by Zahavy et al. (2023) with a VdW diversity objective and a KL-divergence constraint (instead of a linear Value constraint). Furthermore, this imitation constraint can be seen as a relaxation of the SMODICE minimization objective of Ma et al. (2022). We claim that our algorithm is a principled way to combine DOMINO and SMODICE.
> > >
> > > I understand that DOMINO is an online RL method designed for learning diverse behaviors. My point was that the authors could easily evaluate a baseline method that combines SMODICE and DOMINO. I believe this would be distinct from the Dual-Force algorithm, as it additionally incorporates Functional Reward Encoding.
> > > Wouldn't it be beneficial to include such a baseline for comparison?
> > >
> > > >Thank you for your constructive feedback. It is important to emphasize that the reported in Fig. 3 pairwise L2 distance between the successor features of the learned skills is a well-established quantitative measure of skill distinctiveness. Furthermore, the comparison of our results with the previous work of Vlastelica et al. (2024) serves as a valid comparison. Notably, our algorithm learns and zero-shot recalls significantly more diverse skills than previous work.
> > >
> > > While the pairwise L2 distance between the successor features is visualized, I do not see a comparison with Vlastelica et al. (2024) in Figure 3, if I am interpreting the figure correctly. Could you clarify how this comparison is presented in Figure 3?
> > > Even if the method by Vlastelica et al. (2024) can only learn five skills, I believe it should still be possible to quantify the pairwise L2 distance and plot the successor features in the same way. Are these included in Figure 3? I am starting to wonder if I might be misunderstanding how to interpret the figure.

---

> > > > ### Author Response · Authors · 2025-03-03
> > > > **Response to Reviewer PR4G**
> > > >
> > > > > I understand that DOMINO is an online RL method designed for learning diverse behaviors. My point was that the authors could easily evaluate a baseline method that combines SMODICE and DOMINO. I believe this would be distinct from the Dual-Force algorithm, as it additionally incorporates Functional Reward Encoding. Wouldn't it be beneficial to include such a baseline for comparison?
> > > >
> > > > Thank you for raising this important question.
> > > > Of course, it is possible to remove the FRE latent representation, and while in this case the method will learn some diverse skills, the current policy checkpoint will only have access to the last three skills, which may not be useful, as noted in Section 5 “Skills Evaluation”.
> > > > Here, usefulness is measured by achieving at least 50% of the expert's optimal return (note that the reward is never revealed to our algorithm), i.e., the skills may fail to imitate the expert, due to intermediate phases that optimize for diversity.
> > > > The reason for this is that the non-stationary ``alternative optimization scheme’’ does not converge to a stable saddle point, but rather oscillates between optimizing diversity or imitation, depending on the current constraint violation and the associated Lagrange multipliers.
> > > >
> > > > It is important to emphasize that the approach without FRE forces us to store the policy NN weights for each checkpoint during execution, which is significantly more expensive than storing only the latent FRE representations, and post-training to load and evaluate each of these policy checkpoints and distill into a single policy all useful skills for downstream tasks.
> > > > In contrast, the proposed method (Dual-Force) learns a single policy checkpoint and stores a compact set of FRE representations, allowing us to recall all learned skills and more easily select those that meet the above imitation criteria.
> > > > In addition, the FRE latent representation serves as a hash-map that allows for more stable learning of the Value function (and policy) since it is now conditioned on each non-stationary reward, thus satisfying the DICE assumption of optimizing a stationary reward.
> > > >
> > > >
> > > > > While the pairwise L2 distance between the successor features is visualized, I do not see a comparison with Vlastelica et al. (2024) in Figure 3, if I am interpreting the figure correctly. Could you clarify how this comparison is presented in Figure 3? Even if the method by Vlastelica et al. (2024) can only learn five skills, I believe it should still be possible to quantify the pairwise L2 distance and plot the successor features in the same way. Are these included in Figure 3? I am starting to wonder if I might be misunderstanding how to interpret the figure.
> > > >
> > > > Thank you for your important comment.
> > > > Although it would be interesting to evaluate the L2 distance between the successor features of the learned DOI skills, in our work we focus on the different aspect that the Dual-Force algorithm significantly increases the cardinality of the set of diverse skills learned while achieving stable training and enjoying a strong diversity signal.
> > > > Furthermore, Zahavy et al [1, Section 6.2 GAIL and DIAYN] gave a geometric interpretation showing that the Mutual Information (MI) objective in DOI provides a weaker-average guarantee, while the VdW objective enjoys a stronger-pairwise (worst-case) guarantee.
> > > > In particular, the MI objective maximizes the KL-divergence between each state-occupancy and the *mean* state-occupancy, effectively placing the skills on the boundary of an ellipsoid without ensuring that each pair of state-occupancies is far from each other!
> > > > In contrast, the VdW objective guarantees that the minimum L2 distance between the successor features of each pair of learned skills is equal to a predefined threshold $\ell_0$.
> > > > In the updated manuscript, see Figure S1 in Appendix E Training Metrics, we have provided empirical evidence that the Dual-Force algorithm achieves the specified VdW threshold $\ell_0$.
> > > >
> > > > [1] T. Zahavy, B. O'Donoghue, G. Desjardins, and S. Singh. Reward is enough for convex MDPs. arXiv preprint arXiv:2106.00661, 2021.

---

> > ### Comment · Reviewer_PR4G · 2025-03-03
> >
> > The part where the contribution was unclear has been resolved by providing the correct reference to https://rlj.cs.umass.edu/2024/papers/RLJ_RLC_2024_169.pdf.
> >
> > I would like to confirm whether I have correctly understood the proposed method. Is it correct to understand that the proposed method consists of the following three elements?
> > 1. The objective function that promotes diversity based on the Van der Waals (VdW) force, as proposed in DOMINO,
> > 2. Density ratio estimation similar to SMODICE,
> > 3. The use of latent variables obtained through Functional Reward Encoding (FRE).
> >
> > I believe that DOI by Vlastelica et al. (2024) consists of two elements:
> > 1. An objective function that promotes diversity using a discriminator,
> > 2. Density ratio estimation from SMODICE.
> >
> > If I have misunderstood the methods, please correct me.
> >
> > I assume that Figures 4 and 5 show the results of the proposed method. Would it be impossible to obtain similar results using DOI by Vlastelica et al. (2024)? If DOI fails to achieve similar results, could you visualize how it fails in the same format as Figures 4 and 5?

---

> > > ### Author Response · Authors · 2025-03-03
> > > **Response to Reviewer PR4G**
> > >
> > > > 1. I would like to confirm whether I have correctly understood the proposed method. Is it correct to understand that the proposed method consists of the following three elements? ...
> > > 2. I assume that Figures 4 and 5 show the results of the proposed method.
> > > 3. Would it be impossible to obtain similar results using DOI by Vlastelica et al. (2024)?
> > > 4. If DOI fails to achieve similar results, could you visualize how it fails in the same format as Figures 4 and 5?
> > >
> > > Thank you for your valuable suggestion.
> > > Indeed, this is an accurate description of the method formulations of DOI and Dual-Force.
> > > This is correct, Figures 4 and 5 show the results of the Dual-Force algorithm.
> > > For Figure 4 in our work, Vlastelica et al. (2024) already presented in Figure S5 in the Appendix of their work three selected skills possibly from the same policy checkpoint.
> > > In Figure 5, instead of comparing with prior work, we use a standard evaluation scheme from the Quality-Diversity (QD) literature (Zahavy et al., 2023) to show that it is beneficial to discover a diverse set of high-performing skills to ensure robustness and adaptability.
> > > However, it is beyond the scope of this work to experimentally reproduce the DOI algorithm of Vlastelica et al. (2024).

---

> > > > ### Comment · Reviewer_PR4G · 2025-03-04
> > > >
> > > > > In Figure 5, instead of comparing with prior work, we use a standard evaluation scheme from the Quality-Diversity (QD) literature (Zahavy et al., 2023) to show that it is beneficial to discover a diverse set of high-performing skills to ensure robustness and adaptability.
> > > >
> > > > I am aware that K-shot adaptation is often used to evaluate the diversity of learned skills, as seen in studies such as Zahavy et al. (2024) and Kumar et al. (2020). However, in these studies, K-shot adaptation performance was compared with other methods to assess diversity. In contrast, the proposed method does not include any comparison with existing approaches regarding adaptation to environmental changes. Without such a comparison, it is difficult to assess the diversity of the skills learned by the proposed method.

---

> > > > > ### Author Response · Authors · 2025-03-06
> > > > > **Response to Reviewer PR4G**
> > > > >
> > > > > > I am not fully convinced why the authors have not visualized the pairwise L2 distance between the successor features for DOI proposed by Vlastelica et al. (2024). Without such a comparison, it is difficult to assess the performance of the proposed method relative to existing approaches. If the authors wish to claim that the proposed method achieves more diverse behaviors than existing approaches, it is essential to include comparative evaluations.
> > > > >
> > > > > We respectfully disagree.
> > > > > As explained in the previous discussion, the DOI algorithm reports only three learned skills for each task, see Figures S4, S5, and S6 in their work.
> > > > > In contrast, we show that the Dual-Force algorithm learns significantly more skills, most of which have noticeably different behaviors. In particular,
> > > > > - for the locomotion task: similar to DOI, it recovers the three base-heights (low, middle, high), but additionally learns different angular velocities for each height, as shown in Figure 2.
> > > > > - for the obstacle navigation task: similar to DOI, it recovers the left, right, and over-the-box trajectories, but in addition it learns different approach angles, velocities, and even interpolates between these 3+2 modalities, as shown in Figure 4.
> > > > >
> > > > > Our claim, which we have empirically validated, is that the number of learned skills is significantly larger than in previous work, and also a large fraction of all pairwise successor feature distances between the learned skills is at least the specified distance $\ell_0$, as depicted in Figure S1, Figure 3, and is also evident from the skill behaviors in Figure 4.
> > > > >
> > > > > > I am aware that K-shot adaptation is often used to evaluate the diversity of learned skills, as seen in studies such as Zahavy et al. (2024) and Kumar et al. (2020). However, in these studies, K-shot adaptation performance was compared with other methods to assess diversity. In contrast, the proposed method does not include any comparison with existing approaches regarding adaptation to environmental changes. Without such a comparison, it is difficult to assess the diversity of the skills learned by the proposed method.
> > > > >
> > > > > Thank you for your important suggestion.
> > > > > However, the motivation behind Figure 5 is not to assess the diversity of skills learned, but rather to provide a simple illustrative example that conveys "learning different skills, each of which satisfies a certain imitation criterion, is beneficial for adapting to environmental changes", since each skill learns a different modality and some of them then remain robust to environmental changes.
> > > > >
> > > > > In summary, we propose and evaluate a novel offline algorithm with several advantageous features.
> > > > > However, due to resource limitations, it is beyond the scope of this work to experimentally reproduce the DOI algorithm of Vlastelica et al. (2024).

---

> ### Comment · Reviewer_PR4G · 2025-03-04
>
> > Thank you for your important comment. Although it would be interesting to evaluate the L2 distance between the successor features of the learned DOI skills, in our work we focus on the different aspect that the Dual-Force algorithm significantly increases the cardinality of the set of diverse skills learned while achieving stable training and enjoying a strong diversity signal.
>
> I am not fully convinced why the authors have not visualized the pairwise L2 distance between the successor features for DOI proposed by Vlastelica et al. (2024). Without such a comparison, it is difficult to assess the performance of the proposed method relative to existing approaches. If the authors wish to claim that the proposed method achieves more diverse behaviors than existing approaches, it is essential to include comparative evaluations.

---

### Review · Reviewer_QNzk · 2025-02-17

**Summary Of Contributions:**

The paper studies offline imitation learning, primarily using the framework introduces in the DICE series of papers. This work in particular focuses on skill discovery: fitting a latent variable $z$, representing different skills in the dataset. When doing skill discovery, the central question is how to make different $z$ cover different behaviors, usually requiring a diversity objective of some kind.

In this paper, the problem is set as a constrained MDP. The goal is to fit a set of policies $\Pi = \\{\pi_i\\}^{n}_{i=1}$ that maximizes diversity, subject to each policy $\pi_i$ staying close to the optimal policy. In an imitation learning setting, we say the policy is optimal if the stationary distribution of the policy $\pi_i$ equals the offline dataset $S$, and a policy is close to the optimal value if the KL divergence is within $\epsilon$ for some hyperparam $\epsilon$.

The distance between policies is measured by the successor features, in particular the L2 distance $||\psi_i - \psi_j||^2$, and physically inspired Van der Waals forces are used to encourage policies to arrange themselves at diverse uniform points in space. This is based on the work of Zahavy et al 2023, and is inspired by how atoms arrange themselves into crystal lattices by a mix of attractive and repulsive forces.

The paper's contribution is observing that the Van der Waals inspired diversity objective can be made compatible with offline imitation learning DICE framework. Along the way, they use Functional Reward Encodings (FREs) from Frans et al 2024 to pretrain a reward embedding. The bulk of the paper is on carrying through the derivation of the update for Van der Waals inspired diversity maximization to give the final reward term that trades off Van der Waals diversity with imitating the expert policy. Particular attention is paid to how the reward is non-stationary due the added diversity terms, and ways to mitigate the instability of the training via FRE.

**Audience:**

Yes

**Claims And Evidence:**

Yes

**Requested Changes:**

N/A

**Strengths And Weaknesses:**

The paper does a good job of citing the component works for the overall reward definition, which is important given that this work is primarily mixing a number of different papers together. Although there is a lot of math in the paper, it is all explained in sufficient detail, given time to work through the equations.

The evaluation of the skill learning is done by demonstrating how well the skill discovery finds different locomotion gaits for a quadraped, with different heights of the base and differing angular velocities.. The baseline used is a SMODICE-expert, but this feels like a somewhat weird baseline. As I understand it, SMODICE just tries to match the expert state occupency, fits a single policy, and does not try to learn a set of policies, much less a diverse set of policies. To me this seems like the strongest criticism of the paper, that it introduces a lot of machinery for better offline skill learning but doesn't seem to compare to other baselines for doing so.

Nevertheless, I do think the work done to unify FREs + Van der Waals diversity + DICE offline imitation is of interest, at minimum to introduce these ideas to those who may not have seen some of the individual papers before.

---

> ### Author Response · Authors · 2025-02-28
> **Response to Reviewer QNzk**
>
> Thank you for recognizing the value of our work and for your insightful suggestions and constructive feedback. We address the issues raised below.
>
> > The baseline used is a SMODICE-expert, but this feels like a somewhat weird baseline. As I understand it, SMODICE just tries to match the expert state occupancy, fits a single policy, and does not try to learn a set of policies, much less a diverse set of policies. To me this seems like the strongest criticism of the paper, that it introduces a lot of machinery for better offline skill learning but doesn't seem to compare to other baselines for doing so.
>
> Thank you for your important comment. However, we believe there is a misunderstanding. In this work, we consider as a baseline the DOI algorithm proposed by Vlastelica et al. (2024). Our experiments closely follow their setup, see Fig. S4 (locomotion) and Fig. 6 (obstacle navigation) in (Vlastelica et al. 2024), and thus serve as a valid comparison to previous work. Moreover, the experimental results presented in Figures 2, 3, and 4 clearly support our main claim that the Dual-Force algorithm learns a significantly larger set of diverse skills and can zero-shot recall all skills encountered during training, in contrast to the DOI algorithm, which learns only the last five skills and forgets all previously explored skills. It is crucial to emphasize that we have provided the SMODICE-expert only to visualize the state-only imitation constraint, but it is the DOI algorithm that we compare against.
>
> In addition, in Fig. 5, we show that the set of learned skills provides a robust solution for reaching the target position even in an adversarial setting where a fence obstacle partially blocks the path to the target position. Following the standard evaluation approach in the Quality-Diversity (QD) literature, see Zahavy et al. (2023) or our response to Reviewer n7FF's comment C2, for each fence configuration, we select the best performing skills and compare them with the SMODICE-expert. This comparison shows that it is beneficial to discover a diverse set of high-performing skills to ensure robustness and adaptability.
>
> We will update the manuscript to reflect and highlight this discussion.

---

### Review · Reviewer_n7FF · 2025-02-20

**Summary Of Contributions:**

This paper presents a diversity maximisation algorithm for the offline setting. It builds on prior work for diversity maximisation using objectives based on successor features and vanderwaals forces. The key contribution is an offline estimator of the van-der-waals forces, removing the need for an online discriminator. By conditioning the policy and value function on the Functional Reward Encoding, they better handle the training of the networks and dealing with non-stationary rewards.

**Audience:**

Yes

**Broader Impact Concerns:**

No major concerns.

**Claims And Evidence:**

Yes

**Requested Changes:**

No major changes. Minor suggestions for improvement for the authors above.

**Strengths And Weaknesses:**

The paper is written clearly, and accomplishes its goals listed in its contributions. Other than the theoretical contributions towards offline RL/optimization and handling non-stationary rewards in this setting, the results support the contributions presented.

Questions and improvements:
- What is the state-space used when computing the successor features and hence used on the vanderwaals forces? Is it just the torso z height and angular velocity of the torso? Is this pre-selected ion purpose? I think this should be more clear in the paper.
- In the related work, I think it is worth acknowledging the work done in quality-diversity algorithms, which is related to unsupervised skill-discovery. Which was also referenced and cited by Zahavy et al.

---

> ### Author Response · Authors · 2025-02-28
> **Response to Reviewer n7FF - Questions and improvements**
>
> Thank you for recognizing the value of our work and for your insightful suggestions and constructive feedback. We address the issues raised below.
>
> > C1. What is the state-space used when computing the successor features and hence used on the Van der Waals forces? Is it just the torso z height and angular velocity of the torso? Is this pre-selected on purpose? I think this should be more clear in the paper.
>
> This is an excellent point. For fair comparison and consistency, our experiments closely follow the setup in Vlastelica et al. (2024). The Dual-Force algorithm is trained on offline datasets, whose collection process is described in Section G "Solo-12 Dataset Collection" of their work.
>
> In particular, for both robotic tasks (locomotion and obstacle navigation), the state-discriminator and the SMODICE-expert are trained on the full state space, forcing the learned skills to imitate the state-only expert demonstrations in their entirety. In contrast, as reported by Vlastelica et al. (2024), it is beneficial in practice to consider a diversity objective with successor features induced by a projection onto the most relevant components of the state space.
>
> In this work, the following design choices are made for the diversity objective:
> - For the locomotion task, we consider a projection onto the “joint positions (dof_pos)” represented by a 12-dimensional vector. This vector describes the movements for each of the four legs - front left, front right, hind left, and hind right: (hip abduction/adduction, hip flexion/extension, and knee flexion/extension). It serves as a proxy for the body height, which is not present in the offline dataset. Interestingly, the learned skills recover the three body heights and different angular velocities.
> - For the obstacle navigation task, we consider a projection onto the “base linear velocity” and “base angular velocity”, each represented by a 3-dimensional vector. This choice encourages diversity, as most trajectories in the offline dataset have similar body heights on the ground, but approach the obstacle from different directions and at different speeds. While many of the learned skills recover specific modalities within the offline dataset, interestingly, there are also many skills that learn mixtures of these modalities.
>
> As a side note, the state space for the locomotion task contains 48-dimensions (base linear velocity 3-dim, base angular velocity 3-dim, joint positions 12-dim, joint velocity 12-dim, projected gravity 3-dim, commanded velocity 3-dim, previous action 12-dim) while the obstacle navigation task contains 171-dimensions (in addition to the base 48-dim, a surrounding height map of the robot 121-dim and a time information 1-dim).
>
> We will update the manuscript to reflect this discussion.
>
> > C2. In the related work, I think it is worth acknowledging the work done in quality-diversity algorithms, which is related to unsupervised skill-discovery. Which was also referenced and cited by Zahavy et al.
>
> Thank you for the valuable suggestion. Following the reviewer's request, we will update the manuscript with the following paragraph in the Related Work section:
>
> Quality-Diversity. Discovering a diverse set of optimal or near-optimal policies can improve the robustness and generalization of RL agents. The literature largely encompasses two main algorithmic families: MAP-Elites [2, 4] and novelty search with local competition [3]. QD algorithms use evolutionary strategies to maintain and adapt a collection of policies, aiming to balance the quality and diversity trade-off [1, 5, 6].
>
>
> [1] Cully, A. Autonomous skill discovery with quality-diversity and unsupervised descriptors. In
> Proceedings of the Genetic and Evolutionary Computation Conference, pp. 81–89, 2019.
>
> [2] Cully, A., Clune, J., Tarapore, D., and Mouret, J.-B. Robots that can adapt like animals. Nature, 521(7553):503–507, 2015
>
> [3] Lehman, J. and Stanley, K. O. Evolving a diversity of virtual creatures through novelty search and local competition. In Proceedings of the 13th annual conference on Genetic and evolutionary computation, pp. 211–218, 2011.
>
> [4] Mouret, J.-B. and Clune, J. Illuminating search spaces by mapping elites. arXiv preprint
> arXiv:1504.04909, 2015.
>
> [5] Pugh, J. K., Soros, L. B., and Stanley, K. O. Quality diversity: A new frontier for evolutionary
> computation. Frontiers in Robotics and AI, 3:40, 2016.
>
> [6] Tarapore, D., Clune, J., Cully, A., and Mouret, J.-B. How do different encodings influence
> the performance of the map-elites algorithm? In Proceedings of the Genetic and Evolutionary
> Computation Conference 2016, pp. 173–180, 2016.

---

### Author Response · Authors · 2025-02-28
**General Authors Response**

We would like to thank the reviewers for the time they spent reviewing our paper and for their insightful comments.

Several reviewers recognized the novelty of the setting studied, the theoretical analysis and the empirical evaluation on the two robotic tasks in simulation. More specifically:

[key contributions] “an offline estimator of the van-der-waals forces”, “removing the need for an online (skill) discriminator”, “conditioning the policy and value function on FRE, better handles the training of the networks and dealing with non-stationary rewards”, “results support the contributions presented” (n7FF); “introduces a lot of machinery for better offline skill learning”, “work done to unify FREs + Van der Waals diversity + DICE offline imitation is of interest” (QNzk); “the proposed method appears solid and well-designed” (PR4G)

[presentation] “paper is written clearly, and accomplishes its goals”,  “theoretical contributions towards offline RL/optimization and handling non-stationary rewards” (n7FF); “a good job of citing the component works”, “math in the paper, is all explained in sufficient detail” (QNzk).

Thank you for the positive feedback!

The main concerns raised relate to the novelty (PR4G) and baseline comparison (PR4G, QNzk) of the proposed method.

We regret that there was a typo in the reference section, this caused unnecessary confusion.
The correct link to the baseline method (DOI) that appeared under a different name in "RLJ|RLC 2024" and to which we compare our work is https://rlj.cs.umass.edu/2024/papers/RLJ_RLC_2024_169.pdf

Below is a list of the key updates in the newly uploaded manuscript, highlighted in purple for easier reading:

1. In the introduction section, the novelty of the proposed method is more clearly stated.
2. Additional references to the DICE framework are added to Section 4.2.
3. In the experiments section:
- We clarify the offline data collection process, which effectively replicates the experimental setup in (Vlastelica et al., 2024), allowing a direct comparison between their algorithm (DOI) and ours (Dual-Force);
- We emphasize our main result: Dual-Force enjoys a strong diversity signal and satisfies the imitation constraint (see Supp. E), recalls all skills encountered during training, and significantly expands the set of diverse skills learned as their number scales with more iterations. In contrast, the DOI algorithm only remembers the last five skills.
- We state and motivate the choice of projection onto the state space used for the diversity objective.
4. We extend the related work section with a discussion of the Quality-Diversity literature and a relevant offline skill discovery method based on skill predictability (Liu et al., 2023 -- CLUE).
5. We update the Bibtex references, especially the DOI work.
6. We extend the Appendix with three sections:
- ``Reproducibility’’ which describes real-time computation (in hours), batch sizes, GPU, etc.
- ``Training Metrics’’ of the state discriminator, functional reward encoding, and Dual-Force algorithm. In particular, we show the constraint satisfaction and the strong diversity signal throughout the execution of our algorithm.
- ``Solo12 State Space'' which describes the state space components of the locomotion and obstacle navigation tasks.

We believe our answers and changes detailed in the individual responses clarify the questions
and address the raised concerns about our work.

---

### Decision · Action_Editor_2knc · 2025-03-31

**Recommendation:** Reject

**Comment:**

ACs highly recommend reflecting the reviewers' comments to make this submission stronger. Unlike a conference with no second-round revision, this is a journal submission, and thus it is highly suggested that the revision should be made following the reviewer's opinion (at least partly).

**Audience:**

This work presents a novel offline algorithm for generating a diversity maximization of a learned skill set without training of a skill discriminator. The topic is important and the proposed algorithm could be useful for much audience on offline RL.

**Claims And Evidence:**

Although the proposed algorithm has potential strong merits, the two reviewers feels that the experimental evidence is not convincing enough, since no skill learning baseline is compared. We understand that the RL experiments may require huge amount of compute resources, but it is authors' responsibility to clearly show the effectiveness of the proposed method over the state-of-the-arts.

**Resubmission Of Major Revision:**

The authors may consider submitting a major revision at a later time.